# Constrained Posterior Sampling: Time Series Generation with Hard Constraints

**Sai Shankar Narasimhan**        **Shubhankar Agarwal**        **Litu Rout**

**Sanjay Shakkottai**        **Sandeep Chinchali**

Chandra Family Department of Electrical and Computer Engineering,
The University of Texas at Austin, Austin, TX, 78712.
{nsaishankar,somi.agarwal,litu.rout,sanjay.shakkottai,sandeepc}@utexas.edu

## Abstract

Generating realistic time series samples is crucial for stress-testing models and protecting user privacy by using synthetic data. In engineering and safety-critical applications, these samples must meet certain hard constraints that are domain-specific or naturally imposed by physics or nature. Consider, for example, generating electricity demand patterns with constraints on peak demand times. This can be used to stress-test the functioning of power grids during adverse weather conditions. Existing approaches for generating constrained time series are either not scalable or degrade sample quality. To address these challenges, we introduce Constrained Posterior Sampling (CPS), a diffusion-based sampling algorithm that aims to project the posterior mean estimate into the constraint set after each denoising update. Notably, CPS scales to a large number of constraints ($\sim 100$) without requiring additional training. We provide theoretical justifications highlighting the impact of our projection step on sampling. Empirically, CPS outperforms state-of-the-art methods in sample quality and similarity to real time series by around 70% and 22%, respectively, on real-world stocks, traffic, and air quality datasets.

## 1   Introduction

Realistic time series generation plays a crucial role in tasks such as "what-if" scenario analysis [1], stress-testing machine learning models [2, 3], and anonymizing private user data [4]. Current approaches leverage state-of-the-art generative models, including Diffusion Models (DMs) [1, 5, 6] and Generative Adversarial Networks (GANs) [7, 8], to sample from time series distributions. Ensuring realism and high fidelity through constraint satisfaction remains a key challenge in domain-specific time series generation. For example, generating daily Open-High-Low-Close (OHLC) chart for stock prices of an S&P 500 company requires the high and low values to bound the opening and closing prices (Figure 1).

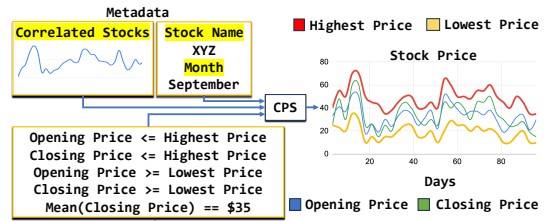

Figure 1: **Our Proposed Constrained Posterior Sampling (CPS).** CPS is a novel diffusion-based sampling approach to generate time series samples that satisfy hard constraints. Here, we show an example where CPS generates a daily stock price time series with natural constraints such as the bounds on the opening and closing prices of the stock.

Similarly, generating stock price time series with user-

39th Conference on Neural Information Processing Systems (NeurIPS 2025).

specified volatility to stress-test trading strategies necessitates adherence to volatility constraints, as inaccuracies in the generated samples lead to inaccurate stress-testing.

Large-scale foundation models have achieved remarkable progress in generative modeling tasks for language and vision [9, 10]. This generative power has sparked significant interest in addressing constraint satisfaction problems [11–15]. However, constraints are often challenging to define and verify in language and vision domains. For instance, verifying the correctness of an image (e.g., a hand with 6 fingers) is inherently difficult due to prediction errors in perception models. In contrast, time series samples can be described through statistical features computed using well-defined functions, and these features can be imposed as constraints and verified accurately. This clarity in defining and verifying constraints makes the time series domain a promising area for advancing research on constraint satisfaction problems. A desired constrained time series generator exhibits the following properties:

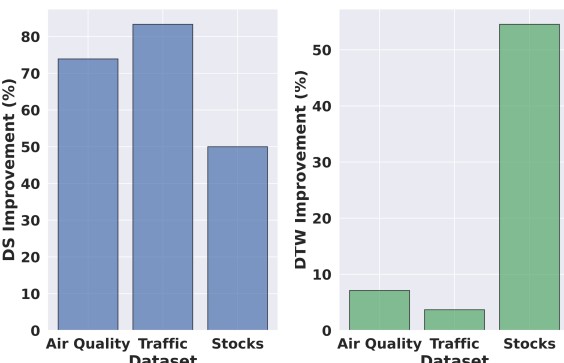

Figure 2: **CPS outperforms the best-performing baselines** on the Dynamic Time Warping (DTW) distance that measures the similarity between the real and the generated samples by 22% and the Discriminative Score (DS), which is the classification error of a model trained to differentiate real and synthetic data, by 70%.

- **Training-free approach to include multiple constraints:** Training a generative model for specific constraints, as in Loss-DiffTime [12], lacks scalability. For example, a model trained for mean constraints cannot easily adapt to argmax constraints.
- **Independence from external realism enforcers:** Prior works often involve projecting samples onto a feasible set defined by constraints, which can degrade sample quality. To mitigate this, previous approaches [12] employ external models alongside the generative model to enforce realism, necessitating additional training and complex sampling procedures.
- **Hyperparameter-free approach to constrained generation:** The choice of guidance weights in guidance-based approaches significantly affects the sample quality. However, optimizing these weights becomes computationally intractable when dealing with hundreds of constraints.

Given these requirements, we propose Constrained Posterior Sampling (CPS), a novel algorithm for diffusion-based time series generation with desired constraints, as illustrated in Figure 1. CPS projects the posterior mean estimate onto the constraint set after each denoising update, leveraging off-the-shelf optimization algorithms and eliminating the need for training or hyperparameter tuning to incorporate multiple constraints. CPS does not require external models to enforce realism, as subsequent denoising steps naturally mitigate the adverse effects of projection. **Our main contributions are as follows:**

1. We present Constrained Posterior Sampling (CPS), a scalable diffusion posterior sampling algorithm to generate time series samples that satisfy desired constraints with high sample quality. CPS efficiently handles a large number of constraints ($\sim 100$) without additional training.
2. We provide rigorous theoretical justification demonstrating the impact of CPS on traditional diffusion sampling. We also provide convergence analysis for well-studied settings (linear constraint sets with Gaussian prior data distribution), offering valuable insights for practice.
3. Through extensive experiments on six diverse real-world and simulated datasets covering finance, traffic, and environmental monitoring, we show that CPS outperforms state-of-the-art methods in sample quality, similarity, and constraint adherence metrics, as shown in Figure 2.

## 2 Preliminaries

**Notations:** We denote a time series sample by $x \in \mathbb{R}^{K \times L}$. Here, $K$ and $L$ refer to the number of channels and the horizon, respectively. A dataset is defined as $\mathcal{D} = \{x^1, \ldots, x^{N_\mathrm{D}}\}$, where the superscript $i \in [1, \ldots, N_\mathrm{D}]$ refers to the sample number, and $N_\mathrm{D}$ is the total number of samples in the dataset. $P_\mathrm{data}$ denotes the real time series data distribution. $x^i$ is the realization of the

random vector $X^i$, where $X^1, \ldots X^{N_D} \sim P_{\text{data}}$. The Probability Density Function (PDF) associated with $P_{\text{data}}$ is represented by $p_{\text{data}} : \mathbb{R}^{K \times L} \to \mathbb{R}$, where $\int p_{\text{data}}(x)dx = 1$. Here, $\int$ refers to the integration operator over $\mathbb{R}^{K \times L}$. The notation $\mathcal{N}(\mu, \Sigma)$ refers to the Gaussian distribution with mean $\mu$ and covariance matrix $\Sigma$. Similarly, $\mathcal{U}(a, b)$ indicates the uniform distribution with non-zero density from $a$ to $b$. $\| \cdot \|_2$ indicates the $l_2$ norm in the case of a vector and the spectral norm in the case of a matrix. We denote the constraint set $\mathcal{C}$ as $\mathcal{C} = \mathcal{C}_1 \bigcap \mathcal{C}_2, \ldots, \bigcap \mathcal{C}_{N_C}$, where $N_C$ is the total number of constraints and $\bigcap$ denotes intersection. Here, $\mathcal{C}_i = \{x \mid f_{c_i}(x) \leq 0\}$ with $f_{c_i} : \mathbb{R}^{K \times L} \to \mathbb{R} \; \forall \; c_i \in [1, \ldots, N_C]$. $\lambda_{\max}(M)$ and $\lambda_{\min}(M)$ refer to the largest and the smallest eigen values of a square matrix $M$ with rank denoted by $rank(M)$.

**Example I:** The stocks dataset has 6 channels ($K = 6$) with 96 timestamps in each channel ($L = 96$). The first 4 channels represent the opening price ($o$), the highest price ($h$), the lowest price ($l$), and the closing price ($c$), and each timestamp represents a day. The OHLC constraint is given by $o - h \leq 0$, $c - h \leq 0$, $l - o \leq 0$, and $l - c \leq 0$, *i.e.,* the opening and closing prices must lie between the highest and the lowest prices. The mean equality constraint on the closing price is expressed as $\frac{1}{L}\left(\sum_{u=1}^{L} c_u\right) - \mu_c \leq 0$ and $\mu_c - \frac{1}{L}\left(\sum_{u=1}^{L} c_u\right) \leq 0$, where $\mu_c$ is the required mean and $c_u$ refers to the value of the closing price for the timestamp $u$.

## 2.1 Diffusion Models

Diffusion models [16, 17] consist of two stochastic processes: (a) forward noising process and (b) reverse denoising process. The **forward process** is modeled by a Markov process where a sample $z_0$ from the data distribution (e.g., a clean image) is transformed into a sample $z_T$ from the standard Gaussian $\mathcal{N}(\mathbf{0}, \mathbf{I})$ in $T$ steps. Let $\mathbf{0}$ and $\mathbf{I}$ denote zero mean and identity covariance. The Markov process uses a Gaussian transition kernel governed by the diffusion coefficients $\bar{\alpha}_0, \ldots, \bar{\alpha}_T$, where $\bar{\alpha}_t \in [0, 1]$ and $\bar{\alpha}_{t-1} > \bar{\alpha}_t \; \forall \; t \in [1, T]$, with boundary conditions given by $\bar{\alpha}_0 = 1, \bar{\alpha}_T = 0$. Formally, $q_t(z_t \mid z_0)$ is the PDF of the conditional Gaussian distribution at time $t$ with mean $\sqrt{\bar{\alpha}_t}z_0$ and covariance matrix $(1 - \bar{\alpha}_t)\mathbf{I}$. The PDF for the marginal at $t = 0$ is given by $q_0 = p_{\text{data}}$.

The **reverse process** transforms a noise sample $z_T \sim \mathcal{N}(\mathbf{0}, \mathbf{I})$ into a clean image from $p_{\text{data}}$. A neural network is trained to align the marginals of the reverse process $p_{\theta,t}(z_{t-1} \mid z_t)$ with that of the forward process for all $t \in [1, T]$. The neural network is usually trained to approximate the mean of $p_{\theta,t}(z_{t-1} \mid z_t)$ for a fixed variance schedule. The resulting training objective becomes:

$$\mathbb{E}_{z_0 \sim P_{\text{data}}, \epsilon \sim \mathcal{N}(\mathbf{0}, \mathbf{I}), t \sim \mathcal{U}(1, T)} [\|\epsilon - \epsilon_\theta(z_t, t)\|_2^2], \tag{1}$$

where $\epsilon_\theta(z_t, t)$ is trained to estimate the noise $\epsilon$ from $z_t$, and $z_t = \sqrt{\bar{\alpha}_t}z_0 + \sqrt{1 - \bar{\alpha}_t}\epsilon$, with $t \sim \mathcal{U}[1, T]$. Interestingly, the training objective (Equation 1) is equivalent to that of a deterministic sampling algorithm, DDIM [18], where the forward noising process is modeled by a non-Markovian process, and the corresponding reverse update becomes:

$$z_{t-1} = \sqrt{\bar{\alpha}_{t-1}}\hat{z}_0(z_t; \epsilon_\theta) + \sqrt{1 - \bar{\alpha}_{t-1} - \sigma_t^2}\epsilon_\theta(z_t, t) + \sigma_t \bar{\epsilon}. \tag{2}$$

Here, $\hat{z}_0(z_t; \epsilon_\theta) = \frac{z_t - \sqrt{1 - \bar{\alpha}_t}\epsilon_\theta(z_t, t)}{\sqrt{\bar{\alpha}_t}}$ is the posterior mean $\mathbb{E}[z_0 | z_t]$, and $\sigma_t$ controls the sampling process stochasticity induced by the noise term $\bar{\epsilon} \sim \mathcal{N}(\mathbf{0}, \mathbf{I})$. From [18], we note that Equation 2 corresponds to the following reverse process:

$$p_{\theta,t}(z_{t-1} \mid z_t) = \begin{cases} p_{\theta,\text{init}}(z_0 \mid \hat{z}_0(z_1; \epsilon_\theta)) & \text{if } t = 1, \\ q_{\sigma,t}(z_{t-1} \mid z_t, \hat{z}_0(z_t; \epsilon_\theta)) & \text{otherwise,} \end{cases} \tag{3}$$

where $q_{\sigma,t}(z_{t-1} \mid z_t, \hat{z}_0(z_t; \epsilon_\theta))$ represents the PDF of the Gaussian distribution with mean $\sqrt{\bar{\alpha}_{t-1}}\hat{z}_0(z_t; \epsilon_\theta) + \sqrt{1 - \bar{\alpha}_{t-1} - \sigma_t^2}\epsilon_\theta(z_t, t)$ and covariance matrix $\sigma_t^2\mathbf{I}$. Similarly, $p_{\theta,\text{init}}(z_0 \mid \hat{z}_0(z_1; \epsilon_\theta))$ is the PDF of the Gaussian distribution with mean $\hat{z}_0(z_1; \epsilon_\theta)$ and covariance matrix $\sigma_1^2\mathbf{I}$. This reverse sampling process can be viewed as obtaining the posterior mean estimate $\hat{z}_0(z_t; \epsilon_\theta)$ and transforming it to the noise level for step $t - 1$. CPS builds on Equation 2.

## 2.2 Related Work

Sampling from distributions constrained to a feasible set is critical in various engineering domains, and prior training-based methods in material science and robotics have addressed this problem.

### 2.2.1 Training-based Constrained Generation

Most training-based solutions [13–15, 19] constrain the sample domain to specific manifolds. This is typically done in prior works with additional trainable layers in Variational Auto Encoders (VAEs) [19] to enforce linear inequality constraints or through constraint-specific training modifications [14, 15], such as clipping the score function for DMs to zero at the constraint boundaries. Recently, Mirror DMs [13] proposed a modified training scheme for convex constraint sets, where standard denoising DMs are trained in the dual or the mirror space of the constraint set. Similarly, recent works [14, 15] proposed a modified forward noising process to restrict the intermediate noisy samples to the constraint set. In the time series domain, Loss-DiffTime [12] proposed a training-based approach where constraint-specific samples are generated by conditioning the generator on the constraints. These approaches lack generalization to new and arbitrary constraint sets, as given in **Example I**.

### 2.2.2 Training-free Constrained Generation

To overcome the limitation of training-based approaches, training-free approaches modify the reverse process either through *projection* or *guidance*. Projected Diffusion Model (PDM) [11] and PRODIGY [20] project the intermediate noisy latents of the reverse process ($z_T, \ldots, z_0$) into the constraint set. Constrained Optimization Problem (COP) [12] projects a seed sample to the constraint set while using the critic function from Wasserstein GAN [21] as a realism enforcer. Although training-free, these approaches adversely impact sample quality and diversity, as shown in Section 5.

Guidance-based approaches [12, 22] push the generated samples toward the constraint set without constraint satisfaction guarantees, even for convex constraints. Similarly, conditional and controlled generation approaches have been extensively studied in the image domain to address tasks such as solving inverse problems [23–26], image personalization [27, 28], text-to-image generation [29, 30], and text-based image editing [31–34]. These methods often involve inversion or proximal gradient updates to guide the reverse process.

Our approach, CPS, belongs to the training-free category, offering improved *efficiency* and sample *quality* with *provable guarantees*. CPS is a proximal gradient descent approach in the same spirit as RB-modulation [27] but with theoretical guarantees for satisfying linear constraints, such as OHLC constraints in stock price generation. We focus on the time series domain, where constraints are defined as statistical features computed using analytical functions, enabling accurate verification.

---

**Problem Setup.** Consider a dataset $\mathcal{D} = \{x^i\}_{i=1}^{N_\mathrm{D}}$, where $N_\mathrm{D}$ denotes the number of samples and $x^i \in \mathbb{R}^{K \times L}$ is the realization of $X^i \sim P_\mathrm{data}$ with the density function $p_\mathrm{data}$. The goal is to generate a typical sample $x^\mathrm{gen} \in \mathbb{R}^{K \times L}$ in the support of $p_\mathrm{data}$ such that $x^\mathrm{gen}$ belongs to the constraint set $\mathcal{C} = \mathcal{C}_1 \bigcap \mathcal{C}_2, \ldots, \bigcap \mathcal{C}_{N_\mathrm{C}}$, where $N_\mathrm{C}$ denotes the number of constraints. Here, $\mathcal{C}_i = \{x \in \mathbb{R}^{K \times L} \mid f_{c_i}(x) \le 0\}$ with $f_{c_i} : \mathbb{R}^{K \times L} \to \mathbb{R} \; \forall \, c_i \in [1, \ldots, N_\mathrm{C}]$.

---

## 3 Method: Constrained Posterior Sampling

In this section, we describe **Algorithm 1** for satisfying the desired constraints while maximizing the likelihood of the generated samples. The procedure alternates between: (1) maximizing likelihood using a pre-trained diffusion model, and (2) enforcing constraints by projecting the expected time series onto the constraint set. In other words, to generate a typical sample in the support of $p_\mathrm{data}$, CPS uses a diffusion model that was trained to maximize the likelihood of observing samples from the given dataset. However, to ensure constraint satisfaction, in CPS, we perturb the denoised sample after every denoising update, using a projection step towards the constraint

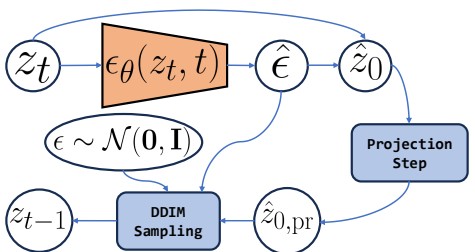

Figure 3: **Constrained Posterior Sampling -** We show the graphical model for one step of denoising in CPS: refer to **Algorithm 1**.

set. Starting with a sample from $\mathcal{N}(\mathbf{0}, \mathbf{I})$ (line 1), we perform sequential denoising using the standard DDIM reverse process (lines 2 to 10). Line 3 refers to the forward pass through the denoiser to obtain the noise estimate $\epsilon_\theta(z_t, t)$. After every denoising step, we obtain the posterior mean estimate $\hat{z}_0(z_t; \epsilon_\theta)$ (line 4) and project it towards the constraint set $\mathcal{C}$ to obtain the projected posterior mean

estimate $\hat{z}_{0,\mathrm{pr}}(z_t; \epsilon_\theta)$ (line 5). Finally, we perform a DDIM reverse sampling step with $\hat{z}_{0,\mathrm{pr}}(z_t; \epsilon_\theta)$ and $\epsilon_\theta(z_t, t)$ to obtain $z_{t-1}$ (lines 7-9).

Figure 3 illustrates one step of the overall pipeline. CPS solves an unconstrained optimization problem in line 5 with the objective function $\frac{1}{2}(\|z - \hat{z}_0(z_t; \epsilon_\theta)\|_2^2 + \gamma(t)\Pi(z))$. The first term of the objective ensures that $\hat{z}_{0,\mathrm{pr}}(z_t; \epsilon_\theta)$ is close to $\hat{z}_0(z_t; \epsilon_\theta)$, ensuring that $z_{t-1}$ remains close to the domain of the denoising network. We define the constraint violation function $\Pi : \mathbb{R}^{K \times L} \to \mathbb{R}$ as $\Pi(z) = \sum_{i=1}^{N_C} \max(0, f_{c_i}(z))$, such that $\Pi(z) = 0$ if $z \in \mathcal{C}$ and $\Pi(z) > 0$ otherwise. For the denoising step $t$, the constraint violation function is scaled by a time-varying penalty $\gamma(t)$.

Our key insight is to design $\gamma(t)$ as a strictly decreasing function of $t$, taking small values for the initial denoising steps and large values for the final denoising steps. This ensures that the constraint satisfaction is not heavily enforced during the initial steps when the signal-to-noise ratio in $z_t$ is very low. We choose $\gamma(t) = e^{1/(1-\bar{\alpha}_{t-1})}$ such that $\gamma(t)$ is close to 0 for the initial steps ($\gamma(T) \approx e$) and $\gamma(t) \to \infty$ for $t = 1$. Here, note that $\gamma(t)$ strictly decreases with $t$ since $\bar{\alpha}_t$ strictly decreases with $t$.

We do not add noise after the final denoising step ($\sigma_1 = 0$), ensuring that the final projection toward constraint satisfaction is not disrupted. For convex constraint sets, the projection step corresponds to an unconstrained minimization of a convex function, with the optimal constraint violation approaching 0 as $\gamma(1) \to \infty$. With an appropriate choice of solvers [35], the optimal solution can be achieved, guaranteeing constraint satisfaction ($\Pi(\hat{z}_{0,\mathrm{pr}}(z_1; \epsilon_\theta)) = 0$) at the end of the reverse process.

CPS meets the key requirements for a constrained generation approach. It handles multiple constraints without additional training or critics to enforce realism, as successive denoising steps naturally correct projection artifacts. CPS introduces no extra hyperparameters, as off-the-shelf solvers [36] handle the unconstrained optimization step (line 5 in **Algorithm** 1). As tuning guidance weights is challenging [12], the tuning-free nature of CPS provides a significant advantage for real-world applications.

CPS is closely related to other training-free constrained generation approaches - PDM [11] and PRODIGY [20]. Unlike CPS which projects $\hat{z}_0(z_t; \epsilon_\theta)$, these approaches project the intermediate latents ($z_T, \ldots, z_0$) onto the constraint set. While ensuring constraint satisfaction

---

**Algorithm 1** Constrained Posterior Sampling

**Input:** Diffusion model $\epsilon_\theta$ with $T$ denoising steps, Noise coefficients $\{\bar{\alpha}_0, \ldots, \bar{\alpha}_T\}$, DDIM control parameters $\{\sigma_1, \ldots, \sigma_T\}$, Constraint violation function $\Pi$, Penalty coefficients $\{\gamma(1), \ldots, \gamma(T)\}$.

1: Initialize $z_T \sim \mathcal{N}(\mathbf{0}, \mathbf{I})$.
2: **for** $t = T$ **to** 1 **do**
3:      Obtain $\hat{\epsilon} = \epsilon_\theta(z_t, t)$
4:      $\hat{z}_0(z_t; \epsilon_\theta) = \frac{z_t - \sqrt{1 - \bar{\alpha}_t}\hat{\epsilon}}{\sqrt{\bar{\alpha}_t}}$
5:      $\hat{z}_{0,\mathrm{pr}}(z_t; \epsilon_\theta) = \arg\min_z \frac{1}{2} \left\{ \begin{array}{l} \|z - \hat{z}_0(z_t; \epsilon_\theta)\|_2^2 \\ + \gamma(t)\Pi(z) \end{array} \right\}$
6:
7:      $z_{t-1} = \sqrt{\bar{\alpha}_{t-1}}\hat{z}_{0,\mathrm{pr}}(z_t; \epsilon_\theta) + \sqrt{1 - \bar{\alpha}_{t-1} - \sigma_t^2}\hat{\epsilon}$
8:      $\epsilon \sim \mathcal{N}(\mathbf{0}, \mathbf{I})$
9:      $z_{t-1} = z_{t-1} + \sigma_t \epsilon$
10: **end for**
11: $x^{\mathrm{gen}} = z_0$
12: **return** $x^{\mathrm{gen}}$

---

for convex sets, these approaches provide poor *sample quality* and *diversity* in comparison to CPS for the following reasons:

- **The constraint set is defined for the clean samples.** However, PDM projects noisy intermediate latents towards the constraint set. This approach eliminates most sample paths ($z_T, \ldots, z_0$) where $z_0$ alone eventually satisfies the constraints, thereby affecting sample diversity. CPS eliminates this problem by projecting $\hat{z}_0(z_t; \epsilon_\theta)$, which has a similar noise level as the constraint set.
- **Projecting $z_{t-1}$ pushes it off the noise manifold for the diffusion step $t - 1$.** Consequently, a pre-trained denoiser struggles to accurately denoise the projected $z_{t-1}$ as it would be out of the training domain of the denoiser. This effect is significantly reduced in CPS because our approach generates $z_{t-1}$ by adding an appropriate amount of noise to the projected clean sample.

To address these issues, PRODIGY adds a scaled version of the projected noisy latent to $z_{t-1}$, such that the projected latent can be given more preference at the final denoising steps. However, due to the direct modification of $z_{t-1}$, PRODIGY still provides lower sample quality and diversity when compared against CPS, as shown in Section 5.

The original DDIM update step results in sampling from an unconditional distribution, whereas updating with $\hat{z}_{0,\mathrm{pr}}(z_t; \epsilon_\theta)$ can be viewed as sampling conditioned on the specified constraints. [27] has employed similar algorithms for image personalization. Note that CPS breaks consistency as $z_t$

is no longer equal to $\sqrt{\bar{\alpha}_t}\hat{z}_{0,\mathrm{pr}}(z_t;\epsilon_\theta) + \sqrt{1-\bar{\alpha}_t}\epsilon_\theta(z_t,t)$ due to the projection step. To understand the effect of this inconsistency, we implemented a variant of CPS called **CPS with noise correction**, where we update the noise estimate to ensure consistency. The adverse effects of the consistency update are shown through poor sample quality metrics in Table 8 (check Appendix B.7). We hypothesize that updating both $\hat{z}_0(z_t;\epsilon_\theta)$ through projection, and $\hat{\epsilon}$ for consistency, can push $z_{t-1}$ off the noise manifold for the step $t-1$, resulting in poor denoising, and thereby affecting the sample quality.

## 4 Theoretical Justification

Now, we provide a detailed analysis of the effect of modifying the DDIM sampling process with CPS. For ease of explanation, we consider $z \in \mathbb{R}^n$. Let $\mathbf{I}_n$ denote the identity matrix in $\mathbb{R}^{n\times n}$. First, we describe the distribution from which the samples are generated under the following assumption.

**Assumption 4.1.** Let the constraint set be $\mathcal{C} = \{z \mid f_\mathcal{C}(z) = 0\}$, where $f_\mathcal{C} : \mathbb{R}^n \to \mathbb{R}$ and the penalty function $\Pi(z) = \|f_\mathcal{C}(z)\|_2^2$ has $L$-Lipschitz continuous gradients, i.e., $\|\nabla\Pi(u)-\nabla\Pi(v)\|_2 \leq L\|u-v\|_2 \ \forall\ u,v \in \mathbb{R}^n$.

**Assumption 4.1** follows prior works [37, 38] on posterior sampling for inverse problems in the image domain. [37] uses linear inverse problems without noise to obtain the reverse sampling step, and replaces the pseudoinverse of the measurement operator with non-linear and non-differentiable operations for reconstructing images compressed using JPEG encoding. [38] provides theoretical guarantees for the proposed sampling algorithms using a linear setting and extends to complex image editing tasks.

**Theorem 4.2.** *Suppose **Assumption 4.1** holds. Given a denoiser $\epsilon_\theta : \mathbb{R}^n \to \mathbb{R}^n$ for a diffusion process with noise coefficients $\bar{\alpha}_0,\ldots,\bar{\alpha}_T$, if $\gamma(t) > 0 \ \forall\ t \in [1,T]$, the denoising step in **Algorithm 1** is equivalent to sampling from the following conditional distribution:*

$$p_{\theta,t}(z_{t-1} \mid z_t) = \begin{cases} p_{\theta,\mathrm{init}}(z_0 \mid \hat{z}_{0,\mathrm{pr}}(z_1;\epsilon_\theta)) & \text{if } t = 1, \\ q_{\sigma,t}(z_{t-1} \mid z_t, \hat{z}_{0,\mathrm{pr}}(z_t;\epsilon_\theta)) & \text{otherwise.} \end{cases}$$

*Here, the PDF of $\mathcal{N}\left(\hat{z}_{0,\mathrm{pr}}(z_1;\epsilon_\theta), \sigma_1^2\mathbf{I}_n\right)$ is represented by $p_{\theta,\mathrm{init}}(z_0 \mid \hat{z}_{0,\mathrm{pr}}(z_1;\epsilon_\theta))$, and the PDF of $\mathcal{N}\left(\sqrt{\bar{\alpha}_{t-1}}\hat{z}_{0,\mathrm{pr}}(z_t;\epsilon_\theta) + \sqrt{1-\bar{\alpha}_{t-1}-\sigma_t^2}\epsilon_\theta(z_t,t), \sigma_t^2\mathbf{I}_n\right)$ by $q_{\sigma,t}(z_{t-1} \mid z_t, \hat{z}_{0,\mathrm{pr}}(z_t;\epsilon_\theta))$. $\sigma_1,\ldots,\sigma_T$ are the DDIM control parameters, and $\gamma(t)$ is the penalty coefficient for the step $t$ in **Algorithm 1**.*

**Implications.** Intuitively, **Algorithm 1** can be viewed as replacing $\hat{z}_0(z_t;\epsilon_\theta)$ with $\hat{z}_{0,\mathrm{pr}}(z_t;\epsilon_\theta)$ in the DDIM sampler. Therefore, the marginal PDFs of the reverse process are obtained by replacing $\hat{z}_0(z_t;\epsilon_\theta)$ with $\hat{z}_{0,\mathrm{pr}}(z_t;\epsilon_\theta)$ in Equation 3. Under the **Assumption 4.1**, the projection step (line 5) becomes a sequence of gradient updates transforming $\hat{z}_0(z_t;\epsilon_\theta)$ to $\hat{z}_{0,\mathrm{pr}}(z_t;\epsilon_\theta)$. Having Lipschitz continuous gradients for $\Pi$ allows for fixed step sizes, which guarantees a reduction in the value of the objective function $\frac{1}{2}(\|z - \hat{z}_0(z_t;\epsilon_\theta)\|_2^2 + \gamma(t)\Pi(z))$ with each gradient update. We refer the readers to Appendix C.1 for the detailed proof.

In general, the theorem draws equivalence between the projection step and sampling from a Dirac delta distribution centered at $\hat{z}_{0,\mathrm{pr}}(z_t;\epsilon_\theta)$. Additionally, **Theorem 4.2** can be extended to non-smooth penalty functions as in **Algorithm 1** because the projection operation (line 5) can be written as a series of updates from $\hat{z}_0(z_t;\epsilon_\theta)$ to $\hat{z}_{0,\mathrm{pr}}(z_t;\epsilon_\theta)$. Therefore, **Theorem 4.2** still provides the updated $p_{\theta,t}(z_{t-1} \mid z_t)$ for **Algorithm 1**. Now, we investigate the convergence properties for **Algorithm 1** under the following assumption.

**Assumption 4.3.** Let the data distribution be $\mathcal{N}(\mu,\mathbf{I}_n)$, where $\mu \in \mathbb{R}^n$, and the constraint set be defined as $\mathcal{C} = \{z \in \mathbb{R}^n \mid Az = y\}$ with $A \in \mathbb{R}^{m\times n}$ such that $rank(A) = n \leq m$. Suppose there exists a unique solution $x^* \in \mathbb{R}^n$ that satisfies the desired constraints in $\mathcal{C}$.

**Assumption 4.3** ensures the existence of a unique solution to the linear problem $Ax = y$. While there exist many methods to solve such problems under this assumption, our focus, as in [24, 39, 40], is to use this problem as a framework to analyze the convergence properties of **Algorithm 1**, providing valuable insights for better practical performance. We note that similar assumptions (unique solution) are made in the theoretical analysis of algorithms designed for sample recovery and image inpainting [39, 40] using diffusion models.

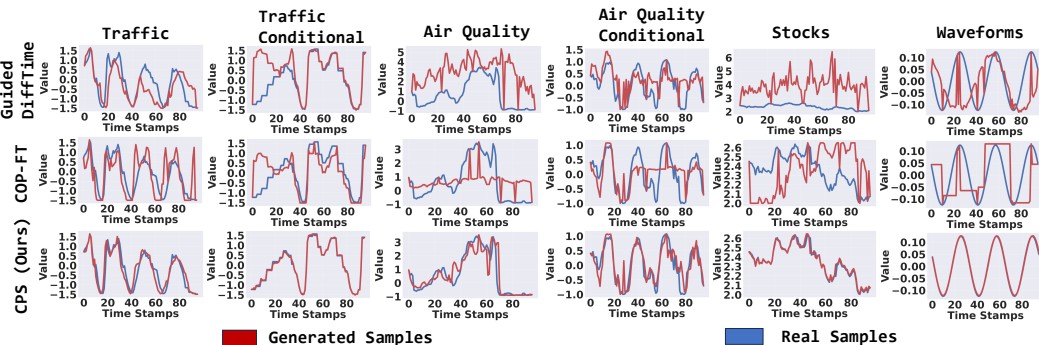

Figure 4: **CPS provides high-fidelity synthetic time series samples that match real time series data.** The real test samples from which the constraints are extracted are shown in blue. The samples generated using the extracted constraints are shown in red. Across all datasets, the baselines suffer from the adversarial effects of the projection step, whereas CPS generates high-quality samples. Here, we provide the comparison against baselines [12] designed for constrained time series generation. We refer the readers to Appendix E, where we included similar qualitative comparisons against other baselines (Diffusion-TS and PDM) that were adapted to perform constrained generation.

**Theorem 4.4.** *Suppose **Assumption 4.3** holds. Let the noise coefficients of a diffusion process be given by $\bar{\alpha}_0, \ldots, \bar{\alpha}_T$, where $\bar{\alpha}_0 = 1$, $\bar{\alpha}_T = 0$ and $\bar{\alpha}_t \in [0,1] \ \forall \ t \in [1,T]$. If $\bar{\alpha}_t < \bar{\alpha}_{t-1}$ and $\gamma(t) = \frac{2k(T-t+1)}{\lambda_{\min}(A^T A)}$, for any $k > 1$, deterministic sampling with **Algorithm 1** converges at the rate $\mathcal{O}(1/T)$. Furthermore, there exists a constant $k > \max\left(1, \frac{\sqrt{\bar{\alpha}_1}}{\delta}\left(\|x^\star\|_2 + \|\mu\|_2\right)\right)$ such that $\|x^{\mathrm{gen}} - x^\star\|_2 \le \delta$ as $T \to \infty$. Here, $\delta$ is the tolerance limit.*

**Implications.** Interestingly, for very large values of $T$, with a suitable choice of $k$, CPS converges without the expensive constrained optimization steps for all denoising steps as required by prior works [11, 20]. Following this insight and the proof in Appendix C.2, we choose the number of denoising steps and the final penalty coefficient to be large enough to ensure convergence in practice. Indeed, it is sufficient to partially enforce the constraints with relatively cheap unconstrained optimization routines for most denoising steps, except for the final step. Intuitively, enforcing constraints partially on the posterior mean estimate, $\hat{z}_0(z_t; \epsilon_\theta)$, noising it back, and repeating this process for a large number of denoising steps removes the adverse effects of projection while gradually nudging the sample generation towards the constraint set.

Overall, **Theorem 4.4** provides a penalty schedule that ensures convergence for **Algorithm 1**. Observe that the penalty coefficient linearly increases to a very large value as we denoise. However, since not all real-world constraints satisfy **Assumption 4.3**, we designed the penalty coefficients in **Algorithm 1** to exponentially increase to a very large value as we denoise, such that they are similar in essence to the penalty coefficients used in **Theorem 4.4**. We conduct extensive experiments in Section 5, verifying the significance of these insights. Specifically, we experimented with linearly and quadratically increasing penalties in **Algorithm 1**, along with exponentially increasing penalty coefficients (check Table 5). Note that the linearly increasing penalty coefficients are proportional to the penalty coefficients obtained from **Theorem 4.4** at all diffusion steps.

## 5 Experiments

In this section, we describe the experiments, datasets, baselines, and metrics used to evaluate CPS.

**Datasets:** We use real-world datasets from the Stocks [7], Air Quality [41], and Traffic [42] domains. We evaluate CPS on both conditional and unconditional variants of these datasets. The conditional variant refers to imposing constraints on conditional generative models that take contextual metadata as input, as mentioned in [1], to generate time series samples. Consequently, in **Algorithm 1**, we use $\epsilon_\theta(z_t, t, m)$ instead of $\epsilon_\theta(z_t, t)$ for the conditional setting, where $m$ is the metadata input. We also evaluate CPS on simulated sinusoidal waveforms with specified amplitudes, phases, and frequencies.

| Metric | Approach | Air Quality | Air Quality (Conditional) | Traffic | Traffic (Conditional) | Stocks | Waveforms |
|---|---|---|---|---|---|---|---|
| Frechet Distance ($\downarrow$) | Guided DiffTime | 0.7457 | 3.1883 | 0.5351 | 0.5638 | 1.2575 | 0.3108 |
| | Diffusion-TS | 0.0473 | 2.9771 | 0.4918 | 0.3561 | 1.1268 | 0.0039 |
| | COP-FT | 0.3821 | 0.9919 | 0.8239 | 0.7836 | 0.0727 | 1.8099 |
| | COP | 0.2206 | 28.1572 | 0.8566 | 43.1499 | 0.0711 | 1.6653 |
| | CPS (Ours) | **0.0234** | **0.6039** | **0.2077** | **0.2812** | **0.0023** | **0.0029** |
| TSTR ($\downarrow$) | Guided DiffTime | 0.29±0.015 | 0.25±0.003 | 0.30±0.01 | **0.28±0.01** | 0.05±0.001 | **0.005±0.001** |
| | Diffusion-TS | **0.19±0.004** | 0.19±0.005 | 0.31±0.01 | **0.28±0.01** | 0.046±0.001 | **0.005±0.001** |
| | COP-FT | 0.23±0.005 | **0.19±0.005** | 0.32±0.01 | **0.28±0.01** | 0.049±0.001 | 0.024±0.001 |
| | COP | 0.25±0.003 | 0.25±0.003 | 0.33±0.02 | 0.32±0.01 | 0.048±0.002 | 0.024±0.002 |
| | CPS (Ours) | **0.19±0.003** | **0.19±0.003** | **0.29±0.01** | **0.28±0.01** | **0.041±0.001** | **0.005±0.001** |
| DS ($\downarrow$) | Guided DiffTime | 0.33±0.02 | 0.22±0.02 | 0.29±0.05 | 0.03±0.02 | 0.38±0.01 | 0.43±0.02 |
| | Diffusion-TS | **0.06±0.03** | 0.03±0.01 | 0.11±0.05 | 0.02±0.01 | 0.21±0.03 | **0.001±0.001** |
| | COP-FT | 0.34±0.03 | 0.03±0.01 | 0.35±0.09 | **0.01±0.01** | 0.13±0.04 | 0.44±0.02 |
| | COP | 0.29±0.02 | 0.30±0.03 | 0.41±0.06 | 0.43±0.01 | 0.09±0.04 | 0.42±0.03 |
| | CPS (Ours) | **0.06±0.01** | **0.01±0.005** | **0.02±0.01** | **0.01±0.004** | **0.006±0.004** | 0.002±0.001 |
| DTW ($\downarrow$) | Guided DiffTime | 6.74±8.18 | 4.28±5.66 | 4.38±1.25 | 1.31±1.01 | 7.84±7.24 | 1.67±1.15 |
| | Diffusion-TS | 2.53±1.96 | 2.30±1.91 | 3.82±1.57 | 0.96±0.52 | 7.44±6.65 | **0.16±0.28** |
| | COP-FT | 3.52±2.08 | 2.01±1.24 | 4.61±1.08 | 1.26±0.88 | 0.90±1.41 | 1.19±0.64 |
| | COP | 4.01±4.71 | 4.13±5.94 | 5.17±1.41 | 4.94±1.08 | 0.88±1.41 | 1.16 ±0.65 |
| | CPS (Ours) | **2.35±1.48** | **1.83±1.16** | **3.41±1.47** | **0.84±0.62** | **0.20±0.71** | 0.23±0.17 |
| Constraint Violation Magnitude ($\downarrow$) | Guided DiffTime | 23.21 | 16.35 | 0.50 | 0.15 | 1128.22 | 5.23 |
| | Diffusion-TS | 5.613 | 3.92 | 0.9743 | 0.45 | 40.51 | 0.36 |
| | COP-FT | **0.0** | **0.0** | **0.0** | **0.0** | **0.0** | 0.0002 |
| | COP | **0.0** | **0.0** | 0.0001 | **0.0** | **0.0** | 0.0003 |
| | CPS (Ours) | **0.0** | **0.0** | **0.0** | **0.0** | **0.0** | **0.0** |

Table 1: **CPS outperforms existing baselines on sample quality and similarity metrics.** The best approach is shown in bold for each metric. Though the guidance-based approaches (Guided DiffTime and Diffusion-TS) provide comparable sample quality metrics, they struggle to provide constraint satisfaction (very high constraint violation magnitudes). Overall, CPS maintains high sample quality (very low FD, TSTR, and DS values) and the highest similarity with real time series samples (lowest DTW). Our key intuition is that the adverse effects of projection are nullified by the subsequent denoising steps. As all constraints are linear, the COP variants and CPS can achieve very low values for constraint violation. The TSTR and DS results are obtained for 3 and 5 seeds, respectively.

**Baselines:** We compare against two post-processing methods - COP [12], which projects a random training sample to the constraint set, and its fine-tuning variant, COP-FT, that projects a generated sample. We also compare against guidance-based methods - **Guided DiffTime** [12] and **Diffusion-TS** [22]. Additionally, we compare against constrained generation approaches from other domains - **Projected Diffusion Models (PDM)** [11] and **PRODIGY** [20]. We use the Time Weaver-CSDI denoiser [1] for all baselines to make a fair comparison.

**Evaluation Procedure:** Our evaluation procedure captures the difference between the real test dataset and the generated dataset with constrained samples, both on distributional and per-sample levels. To achieve this, first, from every real sample in the test dataset, we extract a diverse set of features for constraints - *mean*, *mean consecutive change*, *argmax*, *argmin*, *value at argmax*, *value at argmin*, *values at timestamps 1, 24, 48, 72, & 96*. We then generate a sample for each set of these features using constrained generation approaches. For an ideal constrained generation approach, these two steps combined should be equivalent to sampling from the test data distribution. Additionally, note that the features are extracted from the test samples and, therefore, our evaluation procedure focuses on constraint sets that were never seen during training. Apart from these constraints, we additionally impose the OHLC constraint for the stocks dataset. For waveforms, we extract the peaks, valleys, and the trend from a peak to its adjacent valley. All constraints can be written in the form $Ax \leq 0$, and projection to such constraint sets is easy. The tolerance limit $\delta$ is set to 0.01.

**Metrics:** To compare the samples generated with constraints to the real test samples on a distribution level, we use the Frechet Time Series Distance (**FTSD**) metric [1, 43] for the unconditional setting (also referred to as **Context-FID** [43]) and the Joint Frechet Time Series Distance (**J-FTSD**) metric [1] for the conditional setting. We indicate both these metrics by the Frechet Distance (FD). We also report the Train on Synthetic and Test on Real (**TSTR**) metric for random imputation with 75% masking and the Discriminative Score (**DS**) [7] for sample quality and diversity. We also compute sample-wise comparison metrics, like the Dynamic Time Warping (**DTW**) [44] metric between a real sample and the sample generated with corresponding constraints. This metric is useful when the number of constraints is large, where any desired constrained generation approach should generate a sample that is similar to the test sample from which the constraints are extracted. For constraint

| METRIC | APPROACH | AIR QUALITY | TRAFFIC | STOCKS | AIR QUALITY CONDITIONAL | TRAFFIC CONDITIONAL |
|---|---|---|---|---|---|---|
| FRECHET DISTANCE ($\downarrow$) | PDM | 0.1503 | 0.2714 | 0.0368 | 0.8606 | 0.5510 |
| | PRODIGY | 0.1646 | 0.2771 | 0.0361 | 0.6259 | 0.5811 |
| | CPS (OURS) | **0.0234** | **0.2077** | **0.0023** | **0.6039** | **0.2812** |
| TSTR ($\downarrow$) | PDM | 0.205±0.005 | **0.29±0.008** | 0.044±0.001 | 0.20±0.006 | **0.28±0.01** |
| | PRODIGY | 0.205±0.007 | **0.29±0.005** | 0.0443±0.002 | **0.19±0.004** | **0.28±0.01** |
| | CPS (OURS) | **0.19±0.003** | **0.29±0.001** | **0.041±0.001** | **0.19±0.003** | **0.28±0.01** |
| DTW ($\downarrow$) | PDM | 2.544±1.96 | 3.547±1.34 | 0.447±1.06 | 1.9±1.82 | 1.04±0.77 |
| | PRODIGY | 2.537±1.87 | 3.596±1.37 | 0.516±1.19 | **1.8±1.17** | 1.07±0.78 |
| | CPS (OURS) | **2.35±1.48** | **3.41±1.47** | **0.2±0.71** | 1.83±1.16 | **0.84±0.62** |

Table 2: **CPS outperforms constrained generation approaches from other domains on sample quality and similarity metrics.** CPS outperforms PDM and PRODIGY, which project the noisy intermediate latents, on real-world datasets, highlighting the effectiveness of projecting the posterior mean estimate. Note that both PDM and PRODIGY provide perfect constraint satisfaction, similar to CPS, for the constraints considered in Section 5.

violation, we report the average constraint violation magnitude. A detailed discussion on the baselines and metrics is provided in Appendix H.3 and Appendix F, respectively. Tables 1 and 2 contain comparisons of these metrics between CPS and other baselines on multiple datasets. As such, we make the following observations about the generated sample quality.

**Post-processing through projection is influenced by the choice of the initial seed or the generated sample**. For conditional settings, in COP-FT, conditional generation provides samples closer to the constraint set for post-processing (projection), whereas COP uses a randomly picked training sample. Therefore, the sample quality degradation due to the projection step is low in COP-FT, when compared to COP, resulting in higher FD values (check Table 1). CPS eliminates these effects through iterative projection and denoising, where the adverse effects of the projection step are nullified by the subsequent denoising steps (check Figure 4). Notably, in the Stocks dataset, CPS provides $30\times$ and $15\times$ reduction in the FD and DS values, respectively (Table 1), over the COP variants.

**Projecting the noisy intermediate latents has adversarial effects on sample quality.** As explained in Section 3, projecting the noisy intermediate latents (PDM and PRODIGY) pushes them off the noise manifold, resulting in inaccurate denoising. This affects the generated sample quality as shown through higher FD and TSTR values in Table 2. CPS circumvents this issue by projecting the posterior mean estimate and adding an appropriate amount of noise to the projected posterior mean estimate to obtain the noisy latents (Figure 3). This approach provides noisy latents that are closer to the training domain of the denoiser, eventually resulting in high sample quality, leading to $6\times$ and $10\times$ reduction in the FD values for the Air Quality and Stocks datasets, respectively (Table 2).

**Optimizing the guidance weights seems practically hard for a large number of constraints.** From Table 1, observe the high constraint violation magnitudes for the guidance-based approaches such as Guided DiffTime and Diffusion-TS. We attribute this to the interaction between gradients for each constraint violation, which, if not scaled appropriately, leads to poor guidance. CPS avoids these issues by utilizing off-the-shelf solvers to handle the projection step, and the adverse effects of projection are nullified by the subsequent denoising steps. This results in $50\times$ and $10\%$ reduction in FD and TSTR, respectively, over guidance-based methods for the Stocks dataset.

**CPS effectively tracks the real test samples that adhere to the same set of constraints.** We denote tracking real test samples as the property to have better similarity scores (lower DTW) with the real sample as the number of constraints increases. In Figure 5, we observe that CPS has the best reduction in the DTW scores as the number of constraints increases. Simultaneously, the sample quality is unaffected or even improves for CPS with increasing constraints (lower FD scores). CPS's performance is consistent across multiple real-world datasets, with up to $55\%$ reduction in the DTW scores for the Stocks dataset.

**CPS can be effectively extended to a general class of constraints.** Specifically, we experimented with the Autocorrelation Function (ACF) at a specific lag as an equality constraint along with the OHLC constraint for the stocks dataset. We provide the results of this experiment in Table 4 in Appendix B.3. Note that out of all approaches, CPS provides the least constraint violation magnitude. Additionally, even though the projection step (line 5, **Algorithm 1**) does not lead to the optimal solution (as the autocorrelation function is non-convex in the sample domain), CPS's sample quality is much better than that of the baselines. This is due to the iterated projection and denoising operations, which significantly reduce the adverse effects of the projection step.

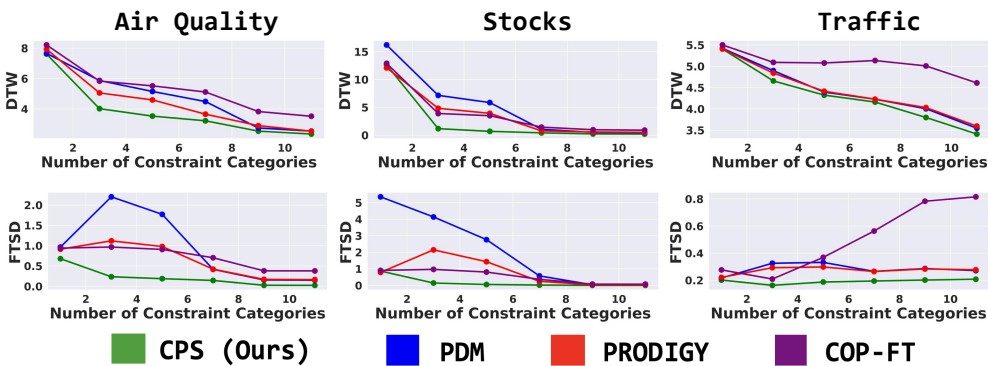

Figure 5: **CPS outperforms baselines on sample quality for an increasing number of constraints.** We gradually increase the number of constraints imposed on the generative model to evaluate CPS and other baselines that guarantee constraint satisfaction for linear constraints. CPS (green) achieves the lowest DTW score for any number of constraints while having the best sample quality, indicated by the lowest FTSD metric. This result is coherent with the qualitative example shown in Figure 6.

**Additional Experiments.** In Appendix B.2, we show that CPS performs as good as training-based approaches, such as Loss DiffTime, on sample quality metrics (Table 3) while ensuring constraint satisfaction. Appendix B.4 (Table 5) provides an ablation study on the choice of the functional form of the penalty coefficients. We also performed a systematic evaluation of the effect of individual constraint categories (Table 6 in Appendix B.5). We note that CPS outperforms the baseline approaches for the majority of these categories. Appendix B.6 contains our detailed experimentation highlighting the effect of guidance weights on the constraint violation magnitude for the guidance-based approaches. In Appendix B.8, we show CPS's ability to outperform other constrained generation approaches in performing time series imputation. Appendix E contains additional qualitative comparisons (Figures 9 and 10).

**Limitations.** Although CPS (**Algorithm 1**) outperforms the compared baselines on standard evaluation metrics, the projection step (line 5) may increase sampling time in some applications, with the trade-off being superior performance. In time-critical scenarios, sampling time can be reduced by leveraging higher-order moments and alternative initialization schemes [25]. Additionally, the projection step does not need to be applied after every denoising step and can be adapted based on the magnitude of constraint violation. Appendix D details the inference time comparisons along with the time complexity analysis for **Algorithm 1** under **Assumptions 4.1** and **4.3**, as well as additional strategies for faster inference. Finally, we note that the convergence results obtained in **Theorem 4.4** hold true under **Assumption 4.3**, which includes restrictions on the data distribution to be Gaussian and the presence of a unique solution. However, as explained in Section 4, these results provide valuable insights for the design of the penalty coefficients used in **Algorithm 1**.

## 6 Conclusion

We proposed Constrained Posterior Sampling (CPS) – a novel training-free approach for constrained time series generation. CPS utilizes off-the-shelf optimization routines to perform a projection step towards the constraint set after every denoising update. In a linear model setting, we provide convergence guarantees for CPS under mild assumptions. Empirically, we show that CPS outperforms the current state-of-the-art in generating realistic samples with superior constraint satisfaction.

**Future work.** We aim to apply CPS for constrained trajectory generation in the robotics domain with dynamic constraints typically modeled by neural networks. Additionally, constrained time series generation readily applies to style transfer applications. Hence, we plan on extending the current work to perform style transfer from one time series to another by perturbing statistical features.

# 7 Acknowledgements

Sai Shankar Narasimhan, Shubhankar Agarwal, and Sandeep Chinchali are part of the Swarm Lab at the University of Texas at Austin.

Litu Rout and Sanjay Shakkottai are supported by NSF Grants 2019844, 2112471, and the UT Austin Machine Learning Lab. Additional Computing support on the Vista GPU Cluster was provided through the Center for Generative AI (CGAI) and the Texas Advanced Computing Center (TACC) at UT Austin.

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

# Appendix

## A   Extended Related Works

### A.1   Diffusion Models for Time Series Generation

Time Series-specific tasks like forecasting [45–49] and imputation [5, 22, 50–52] have been addressed using conditional DMs as well as guidance-based approaches [22, 49]. Specifically, [49, 51] propose novel sampling techniques for improved time series prediction for forecasting and imputation. [6] and [1] have explored conditional time series generation for various domains, such as medical, energy, etc. These works aim to sample from a conditional distribution. Additionally, applications in analogous domains, such as audio [53] and radio frequency waveform generation [54], have also widely adopted diffusion generative modeling, with similar denoiser architectures as seen in the time series domain. However, there are limited prior works in the time series domain [12] that focus on generating constrained samples, which is the focus of this work.

## B   Additional Results

In this section, we provide additional qualitative and quantitative results for the real-world datasets used in our experiments.

### B.1   Effects of increasing the number of constraints

Here, we provide a qualitative example from the Stocks dataset showing CPS's ability to track the real time series sample as the number of constraints is gradually increased.

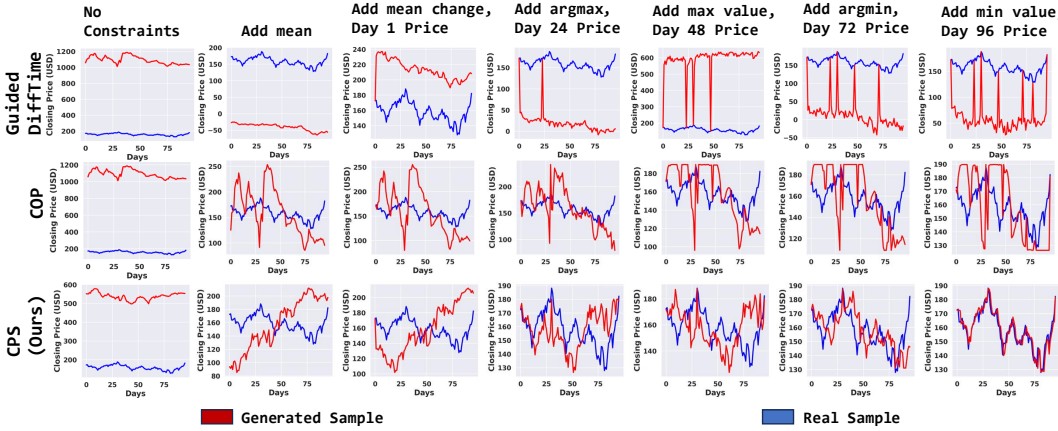

Figure 6: **CPS tracks the real data samples as the number of constraints increases.** Increasing the number of constraints reduces the size of the constraint set, and an ideal approach should effectively generate samples that resemble the real time series samples that belong to the constraint set. Here, we show a qualitative example from the Stocks dataset. Observe that CPS accurately tracks the real sample that concurs with the specified constraints, while other approaches suffer.

### B.2   Extended Baseline Comparisons

In this section, we provide comparisons against the Loss-DiffTime baseline from [12]. For a fair comparison, we use the same TIME WEAVER-CSDI backbone and train the denoiser with constraints as the condition input. The quantitative comparisons are provided in Table 3. As observed with prior approaches, in the absence of any principled projection step, the Loss-DiffTime approach fails to generate samples that adhere to hard constraints. However, due to the constraint-specific training, Loss-DiffTime performs as good as CPS in terms of sample quality and similarity.

| Dataset | Approach | FTSD (↓) | TSTR (↓) | DS (↓) | DTW (↓) | Constraint Violation (↓) Magnitude |
|---|---|---|---|---|---|---|
| Air Quality | Loss DiffTime | **0.0137** | **0.187±0.003** | **0.03±0.01** | **2.18±1.48** | 9.779 |
|  | CPS (Ours) | 0.0234 | 0.19±0.003 | 0.06±0.01 | 2.35±1.48 | **0.0** |
| Stocks | Loss DiffTime | 0.9897 | 0.045±0.002 | 0.379±0.015 | 7.75±6.05 | 237.492 |
|  | CPS (Ours) | **0.0023** | **0.041±0.001** | **0.006±0.004** | **0.20±0.71** | **0.0** |
| Traffic | Loss DiffTime | 0.3653 | **0.29±0.01** | 0.113±0.039 | **3.15±1.34** | 2.993 |
|  | CPS (Ours) | **0.2077** | **0.29±0.001** | **0.02±0.01** | 3.41±1.47 | **0.0** |

Table 3: **Despite constraint-specific training, Loss-DiffTime struggles to generate samples that adhere to the required constraint set.** Note that Loss-DiffTime performs better than CPS on the sample quality and similarity metrics for the air quality dataset. However, due to the absence of projection steps, Loss-DiffTime fails to generate samples that adhere to hard constraints.

## B.3 General Constraints Experiments

We extended our experimental setup to generic constraints for the stocks dataset. Specifically, we imposed the Autocorrelation Function (ACF) at a specific lag as an equality constraint with an acceptable tolerance of 0.01. ACF at a specific lag $l$ for a univariate time series X of horizon $L$ is given by,

$$ACF(X) = \frac{1}{(L-l)\sigma^2} \sum_{u=1}^{L-l} (X_u - \mu)(X_{u+l} - \mu), \tag{4}$$

where $\mu = \mathbb{E}(X)$ and $\sigma^2 = \mathbb{E}[(X - \mu)^2]$, with $\mathbb{E}$ being the expectation operator. Here, $X_u$ and $X_{u+l}$ denote the time series values at the timestamps $u$ and $u + l$, respectively. Note that $\mu$ and $\sigma$ are not fixed. Along with the ACF equality constraint, we pose the OHLC constraint for the stocks dataset. We provide the results of this experiment in Table 4. We chose ACF as it is one of the most popularly used techniques to extract the most relevant lag features for downstream tasks like forecasting.

| Approach | FTSD (↓) | DTW (↓) | Constraint Violation (↓) Magnitude |
|---|---|---|---|
| Guided-DiffTime | 1.4678 | 15.06±11.92 | 284.58 |
| COP | 2.1949 | 72.11±35.97 | 0.9045 |
| CPS (Ours) | **0.0014** | **0.11±0.10** | **0.01** |

Table 4: **CPS outperforms baselines for OHLC and autocorrelation function value constraints.** Here, we use the stocks dataset and impose the Autocorrelation Function (ACF) value for a specified lag of 12 timestamps as a constraint along with the OHLC constraint. CPS outperforms all the baselines in terms of sample quality, similarity, and constraint satisfaction metrics.

Note that out of all approaches, CPS provides the least constraint violation magnitude. Additionally, even though the projection step (line 5, **Algorithm 1**) does not lead to the optimal solution (as the autocorrelation function is non-convex in the sample domain), CPS's sample quality is much better than that of the baselines. We hypothesize that this effect is due to the iterated projection and denoising operations, which significantly reduce the adverse effects of the projection step.

## B.4 Choice of Penalty Coefficients

Our choice of $\gamma(t)$ can take any functional form as long as $\gamma(t) \to \infty$ as $t \to 1$. This is to ensure constraint satisfaction for linear and convex constraint sets. In practice, we clip $\gamma(t)$ to a very large value, such as $10^5$, when performing the final denoising steps. Our current choice of $\gamma(t)$ decreases exponentially with $t$ (in other words, $\gamma(t)$ increases as we denoise). We also experimented with linearly and quadratically decreasing values of $\gamma(t)$, with a very high value ($10^5$) for $t = 1$. We noted that the choice of $\gamma(t)$ has very little effect on the sample quality of the generated samples, with differences in the third decimal (check Table 5).

| Choice of $\gamma(t)$ | Air Quality | Traffic | Stocks |
|---|---|---|---|
| Linear | **0.0222** | 0.2053 | **0.0013** |
| Quadratic | 0.0226 | **0.2027** | 0.0016 |
| Exponential | 0.0234 | 0.2077 | 0.0023 |

Table 5: **Different choices of $\gamma(t)$ provide similar sample quality metrics.** Here, we report the FTSD score as the sample quality metric. Note that the effect of different choices of $\gamma(t)$ is only reflected in the third decimal and is insignificant.

## B.5 Systematic Evaluation Of Different Constraint Categories

We provide the FTSD (sample quality) comparison for individual constraint categories evaluated on the traffic dataset (check Table 6). Note that CPS outperforms the baseline approaches for most of the individual constraint categories.

| Approach | Argmax | Argmin | Value at Argmax | Value at Argmin | Mean | Mean Change | Value at Timestamps 1,24,48,72,96 |
|---|---|---|---|---|---|---|---|
| Guided-DiffTime | 0.21 | 0.22 | 0.21 | 0.30 | 0.22 | 0.22 | 0.24 |
| COP-FT | 0.27 | 0.25 | 0.20 | 0.26 | 0.28 | **0.19** | 0.23 |
| PDM | 0.21 | **0.20** | **0.19** | 0.23 | 0.19 | 0.20 | 0.20 |
| CPS (Ours) | **0.20** | 0.21 | **0.19** | **0.21** | **0.18** | 0.20 | 0.19 |

Table 6: **CPS predominantly outperforms other baselines on sample quality metrics while being evaluated on individual constraints.** The table provides a constraint-specific evaluation of the sample quality (FTSD, lower is better). Here, we experimented with the traffic dataset. Note that for the majority of the constraints, CPS outperforms existing baselines.

## B.6 Effect Of The Guidance Weight On The Constraint Violation Magnitude

We analyze the effect of the guidance weight on the constraint violation magnitude for the Air Quality dataset. Specifically, we experimented with the following guidance weights: $0.00001, 0.00005, 0.0001, 0.0005, 0.001, 0.005, 0.01,$ and $0.05$. We refer the readers to Table 7 for the results. Overall, we observe that the constraint violation magnitude either stays the same (DiffusionTS) or increases (Guided DiffTime) with increasing guidance weight. In our experiments, we choose the guidance weight corresponding to the smallest constraint violation magnitude.

| Approach | Guidance Weight = 0.00001 | Guidance Weight = 0.00005 | Guidance Weight = 0.0001 | Guidance Weight = 0.0005 | Guidance Weight = 0.001 | Guidance Weight = 0.005 | Guidance Weight = 0.01 | Guidance Weight = 0.05 |
|---|---|---|---|---|---|---|---|---|
| Guided DiffTime | 23.21 | 23.94 | 25.37 | 51.27 | 113.31 | 951.62 | 2929 | 116084 |
| Diffusion-TS | 5.61 | 5.67 | 5.68 | 5.68 | 5.7 | 5.67 | 5.64 | 5.67 |

Table 7: **Effect of the guidance weight on the constraint violation magnitude for the guidance-based approaches.** Notice that for both Guided DiffTime and Diffusion-TS, increasing the guidance weight does not result in the reduction of the constraint violation magnitude. The experiments were conducted for the Air Quality dataset.

## B.7 Discussion On The Update Step Consistency

To validate our update step empirically, we implemented a variant of CPS where we update the noise estimate based on $\hat{z}_{0,\text{pr}}(z_t; \epsilon_\theta)$ (check Section 3). This updated noise estimate is used in line 7 of **Algorithm 1**. We refer to this as **CPS with noise correction**. The sample quality results, in the presence and absence of noise correction, are provided in Table 8. Note that updating for consistency significantly affects the sample quality. We hypothesize that updating both $\hat{z}_0(z_t; \epsilon_\theta)$ and $\hat{\epsilon}$ can push $z_{t-1}$ off the noise manifold for the step $t-1$, resulting in poor denoising, and thereby affecting the sample quality.

## B.8 Extending Constrained Posterior Sampling To Time Series Imputation

Here, we extend CPS to the time series imputation task by imposing the **value at** constraint to all the timestamps where the true values are available. We compare CPS against other state-of-the-art

| METRICS | APPROACH | AIR QUALITY | TRAFFIC | STOCKS | WAVEFORMS |
|---------|----------|-------------|---------|--------|-----------|
| FTSD | CPS | **0.0234** | **0.2077** | **0.0023** | **0.0029** |
| | CPS WITH NOISE CORRECTION | 0.2085 | 0.5147 | 0.0426 | 0.0694 |
| DTW | CPS | **2.35 ± 1.48** | **1.83 ± 1.16** | **0.20 ± 0.71** | **0.23 ± 0.17** |
| | CPS WITH NOISE CORRECTION | 3.07 ± 1.86 | 4.05 ± 1.26 | 0.54 ± 1.10 | 0.46 ± 0.33 |

Table 8: **CPS provides better sample quality metrics in the absence of the noise correction step.** Updating both posterior mean and noise (CPS with noise correction) results in overdependency on the projection step and pushes $z_{t-1}$ off the noise manifold for the denoising step $t-1$. This takes $z_{t-1}$ far away from the training domain of the denoiser, resulting in poor sample quality as shown by the numbers above.

constrained generation methods - PDM [11] and PRODIGY [20]. The results are provided in Figure 7. Note that CPS outperforms the baselines on all real-world datasets. This result further cements our claim that projecting the posterior mean estimate, instead of the intermediate latents as in the case of PDM and PRODIGY, is better for constrained sample generation.

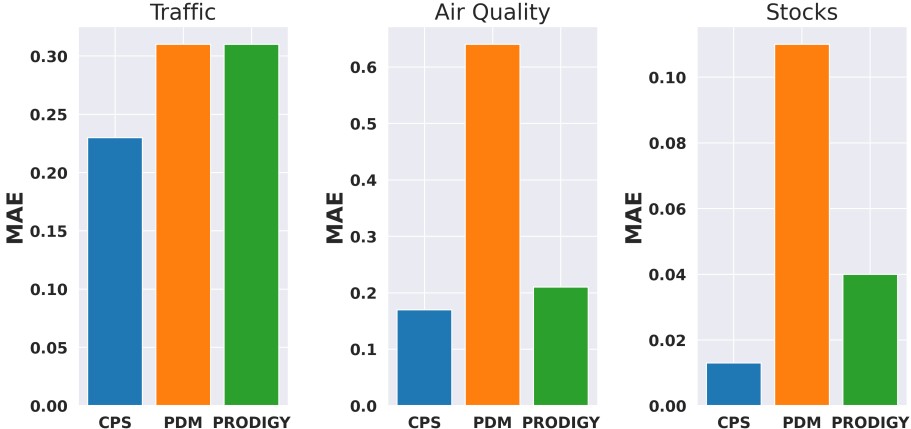

Figure 7: **CPS outperforms state-of-the-art approaches on imputation tasks.** Here, we showcase CPS's ability to perform imputation. We set the masking rate at 50%, *i.e.,* 50% of the values are masked. Observe that in all three real-world datasets, CPS comfortably outperforms PDM [11] and PRODIGY [20] (lowest Mean Absolute Error (MAE)). Note that the main difference between these approaches and CPS is that CPS projects the posterior mean estimate, whereas these approaches project the intermediate noisy latents.

## C  Proofs

In this section, we provide detailed proof for the theorems stated in the manuscript.

### C.1  Proof of Theorem 1

We first describe the assumption on the constraint set. The constraint set is defined as $\mathcal{C} = \{z \mid f_{\mathcal{C}}(z) = 0\}$, where $f_{\mathcal{C}} : \mathbb{R}^n \to \mathbb{R}$, and the penalty function $\Pi(z) = \|f_{\mathcal{C}}(z)\|_2^2$ has $L$-Lipschitz continuous gradients, *i.e.,* $\|\nabla\Pi(u) - \nabla\Pi(v)\|_2 \le L\|u - v\|_2 \; \forall\, u, v \in \mathbb{R}^n$.

Line 7 of the **Algorithm 1** modifies the traditional DDIM sampling by replacing $\hat{z}_0(z_t; \epsilon_\theta)$ with $\hat{z}_{0,\mathrm{pr}}(z_t; \epsilon_\theta)$. Without this modification, the DDIM sampling denotes the following reverse process when started with $x_T \sim \mathcal{N}(\mathbf{0}_n, \mathbf{I}_n)$, where $\mathbf{0}_n$ indicates the zero mean vector in $\mathbb{R}^n$ and $\mathbf{I}_n$ is the identity matrix in $\mathbb{R}^{n \times n}$:

$$p_{\theta,t}(z_{t-1} \mid z_t) = \begin{cases} p_{\theta,\mathrm{init}}(z_0 \mid \hat{z}_0(z_1; \epsilon_\theta)) & \text{if } t = 1, \\ q_{\sigma,t}(z_{t-1} \mid z_t, \hat{z}_0(z_t; \epsilon_\theta)) & \text{otherwise,} \end{cases} \tag{5}$$

where $q_{\sigma,t}(z_{t-1} \mid z_t, \hat{z}_0(z_t; \epsilon_\theta))$ represents the PDF of the Gaussian distribution $\mathcal{N}\left(\sqrt{\bar{\alpha}_{t-1}}\hat{z}_0(z_t; \epsilon_\theta) + \sqrt{1 - \bar{\alpha}_{t-1} - \sigma_t^2}\epsilon_\theta(z_t, t), \sigma_t^2 \mathbf{I}_n\right)$ with $\sigma_t$ as the DDIM control parameter. Similarly, $p_{\theta,\text{init}}(z_0 \mid \hat{z}_0(z_1; \epsilon_\theta))$ is the PDF of the Gaussian distribution with mean $\hat{z}_0(z_1; \epsilon_\theta)$ and covariance matrix $\sigma_1^2 \mathbf{I}_n$ [18].

Note that sampling from $q_{\sigma,t}(z_{t-1} \mid z_t, \hat{z}_0(z_t; \epsilon_\theta))$ provides the DDIM sampling step (check Equation 2).

We reiterate that the main modification with respect to the DDIM sampling approach is the projection step in line 5 of **Algorithm 1**. Therefore, we first analyze the projection step,

$$\hat{z}_{0,\text{pr}}(z_t; \epsilon_\theta) = \arg\min_z \frac{1}{2}\left(\|z - \hat{z}_0(z_t; \epsilon_\theta)\|_2^2 + \gamma(t)\|f_\mathcal{C}(z)\|_2^2\right). \tag{6}$$

Here, $\hat{z}_0(z_t; \epsilon_\theta) = \frac{z_t - \sqrt{1 - \bar{\alpha}_t}\epsilon_\theta(z_t, t)}{\sqrt{\bar{\alpha}_t}}$ (line 3, predicted $z_0$). We will denote the objective function $\frac{1}{2}\left(\|z - \hat{z}_0(z_t; \epsilon_\theta)\|_2^2 + \gamma(t)\|f_\mathcal{C}(z)\|_2^2\right)$ as $g(z)$. Note that we replaced the constraint violation function $\Pi(z)$ by $\|f_\mathcal{C}(z)\|_2^2$ for this case. Given that $\|f_\mathcal{C}\|_2^2$ has $L$-Lipschitz continuous gradients, Equation 6 can be written as a series of gradient updates with a suitable step size such that the value of the objective function decreases for each gradient update.

From the statement, we observe that $\gamma(t) > 0 \; \forall \; t \in [1, T]$. Under this condition and Assumption 4.1, note that the function $g(z)$ is convex and has $\left(\frac{2 + \gamma(t)L}{2}\right)$-Lipschitz continuous gradients, as $\|z - \hat{z}_0(z_t; \epsilon_\theta)\|_2^2$ has 2-Lipschitz continuous gradients, $\gamma(t)\|f_\mathcal{C}(z)\|_2^2$ has $(\gamma(t)L)$-Lipschitz continuous gradients, and the fraction $\frac{1}{2}$ makes $g(z)$ to have $\left(\frac{2 + \gamma(t)L}{2}\right)$-Lipschitz continuous gradients. Let $\eta$ be the step size of the projection step. From [36], we know that $\eta \in (0, 2/(2 + \gamma(t)L))$ ensures that the objective function in Equation 6 reduces after each gradient update. We denote the gradient update as:

$$^n\hat{z}_0(z_t; \epsilon_\theta) = \; ^{n-1}\hat{z}_0(z_t; \epsilon_\theta) - \eta\nabla_z(g(z))\big|_{^{n-1}\hat{z}_0(z_t;\epsilon_\theta)}, \tag{7}$$

where $^0\hat{z}_0(z_t; \epsilon_\theta) = \hat{z}_0(z_t; \epsilon_\theta)$ and $\hat{z}_{0,\text{pr}}(z_t; \epsilon_\theta) = \; ^{N_{\text{pr}}}\hat{z}_0(z_t; \epsilon_\theta)$. Here, $N_{\text{pr}}$ is the total number of gradient update steps. The iteration in Equation 7 always leads to $\hat{z}_{0,\text{pr}}(z_t; \epsilon_\theta)$ deterministically. Therefore, the projection step can be considered sampling from a Dirac delta distribution centered at $\hat{z}_{0,\text{pr}}(z_t; \epsilon_\theta)$, *i.e.*, $\delta(z - \hat{z}_{0,\text{pr}}(z_t; \epsilon_\theta))$. Consequently, using the law of total probability, the reverse process corresponding to the denoising step $t \; \forall \; t \in [2, T]$ in **Algorithm 1** is given by

$$p_{\theta,t}(z_{t-1} \mid z_t) = \int p_{\theta,t}(z_{t-1}, \hat{z}_0 \mid z_t)d\hat{z}_0,$$

where $\hat{z}_0 \in \mathbb{R}^n$. This can be simplified using Bayes' rule as

$$p_{\theta,t}(z_{t-1} \mid z_t) = \int \delta(\hat{z}_0 - \hat{z}_{0,\text{pr}}(z_t; \epsilon_\theta))q_{\sigma,t}(z_{t-1} \mid z_t, \hat{z}_0)d\hat{z}_0.$$

The above equation stems from the fact that the distribution of $z_0$ conditioned on $z_t$ is a Dirac delta distribution centered at $\hat{z}_{0,\text{pr}}(z_t; \epsilon_\theta)$. Since $\delta(x - y) = \delta(y - x)$ and using the sifting property of a Dirac delta function $\left(\int f(z)\delta(a - z)dz = f(a)\right)$, we get

$$p_{\theta,t}(z_{t-1} \mid z_t) = q_{\sigma,t}(z_{t-1} \mid z_t, \hat{z}_{0,\text{pr}}(z_t; \epsilon_\theta)) \; \forall \; t \; \in \; [2, T]. \tag{8}$$

Similarly, we repeat the steps for $t = 1$,

$$p_{\theta,1}(z_0 \mid z_1) = \int p_{\theta,1}(z_0, \hat{z}_0 \mid z_t)d\hat{z}_0,$$

$$p_{\theta,1}(z_0 \mid z_1) = \int \delta(\hat{z}_0 - \hat{z}_{0,\text{pr}}(z_1; \epsilon_\theta))p_{\theta,\text{init}}(z_0 \mid \hat{z}_0)d\hat{z}_0,$$

$$p_{\theta,1}(z_0 \mid z_1) = p_{\theta,\text{init}}(z_0 \mid \hat{z}_{0,\text{pr}}(z_1; \epsilon_\theta)).$$

Combining the two, we get

$$p_{\theta,t}(z_{t-1} \mid z_t) = \begin{cases} p_{\theta,\text{init}}(z_0 \mid \hat{z}_{0,\text{pr}}(z_1; \epsilon_\theta)) & \text{if } t = 1, \\ q_{\sigma,t}(z_{t-1} \mid z_t, \hat{z}_{0,\text{pr}}(z_t; \epsilon_\theta)) & \text{otherwise,} \end{cases} \tag{9}$$

where $q_{\sigma,t}(z_{t-1} \mid z_t, \hat{z}_{0,\mathrm{pr}}(z_t; \epsilon_\theta))$ represents the PDF of the Gaussian distribution $\mathcal{N}(\sqrt{\bar{\alpha}_{t-1}}\hat{z}_{0,\mathrm{pr}}(z_t; \epsilon_\theta) + \sqrt{1 - \bar{\alpha}_{t-1} - \sigma_t^2}\epsilon_\theta(z_t, t), \sigma_t^2 \mathbf{I}_n)$ with $\sigma_t$ as the DDIM control parameter. Similarly, $p_{\theta,\mathrm{init}}(z_0 \mid \hat{z}_0(z_1; \epsilon_\theta))$ is the PDF of the Gaussian distribution with mean $\hat{z}_{0,\mathrm{pr}}(z_1; \epsilon_\theta)$ and covariance matrix $\sigma_1^2 \mathbf{I}_n$ [18].

We note that the value of $\sigma_1$ is set to 0 in **Algorithm 1**. However, similar to [18], for theoretical analysis, we consider a negligible value for $\sigma_1$ ($\sim 10^{-12}$) to ensure that the generative process is supported everywhere. In other words, $\sigma_1$ is chosen to be so low such that for $\sigma_1 \simeq 0$, $p_{\theta,\mathrm{init}}(z_0 \mid \hat{z}_{0,\mathrm{pr}}(z_1; \epsilon_\theta)) \simeq \delta(z_0 - \hat{z}_{0,\mathrm{pr}}(z_1; \epsilon_\theta))$.

Now, we show that the exact DDIM reverse process (check Equation 5) can be obtained from Equation 9 in the case where there are no constraints. Here, note that in the absence of any constraint, the projection step can be written as $\hat{z}_{0,\mathrm{pr}}(z_t; \epsilon_\theta) = \arg\min_z \frac{1}{2}\|z - \hat{z}_0(z_t; \epsilon_\theta)\|_2^2$, in which case $\hat{z}_{0,\mathrm{pr}}(z_t; \epsilon_\theta) = \hat{z}_0(z_t; \epsilon_\theta)$.

For $t \in [2, T]$, using the law of total probability, we get

$$p_{\theta,t}(z_{t-1} \mid z_t) = \int \delta(\hat{z}_0 - \hat{z}_0(z_t; \epsilon_\theta))q_{\sigma,t}(z_{t-1} \mid z_t, \hat{z}_0)d\hat{z}_0, \tag{10}$$

which simplifies further to

$$p_{\theta,t}(z_{t-1} \mid z_t) = q_{\sigma,t}(z_{t-1} \mid z_t, \hat{z}_0(z_t; \epsilon_\theta)). \tag{11}$$

The above equation stems from the same sifting property of Dirac delta functions. The same applies to $t = 1$, except that after the projection step since there is no necessity for constraint satisfaction, we sample from $p_{\theta,\mathrm{init}}(z_0 \mid \hat{z}_0(z_1; \epsilon_\theta))$, which is a Gaussian distribution with mean $\hat{z}_0(z_1; \epsilon_\theta)$ and covariance matrix $\sigma_1^2 \mathbf{I}_n$. Combining both cases, we observe that without any constraints the exact DDIM reverse process can be recovered from **Algorithm 1** for all $t \in [1, T]$.

### C.2 Proof of Theorem 2

We note that the intermediate samples in a $T$-step reverse sampling process are denoted as $z_T, \ldots, z_0$, where $z_0 = x^{\mathrm{gen}}$ and $z_T \sim \mathcal{N}(\mathbf{0}_n, \mathbf{I}_n)$. Once again, we reiterate the assumptions. We consider the real data distribution to be Gaussian with mean $\mu \in \mathbb{R}^n$ and covariance matrix $\mathbf{I}_n$, *i.e.*, $\mathcal{N}(\mu, \mathbf{I}_n)$. The constraint set $\mathcal{C}$ is defined as $\mathcal{C} = \{z \mid Az = b\}$ with $A \in \mathbb{R}^{m \times n}$ such that $rank(A) = n$, where $m \geq n$. Additionally, for the real data distribution $\mathcal{N}(\mu, \mathbf{I}_n)$ and the constraint set $\mathcal{C} = \{z \mid Az = y\}$, there exists a unique sample $x^* \in \mathbb{R}^n$ that satisfies the desired constraints in $\mathcal{C}$. We build on the problem setting presented in prior works [24, 39, 40] to understand the implications of our constrained sampling algorithm. While the prior works focus on a DDPM-based sampler, our analysis focuses on constraint satisfaction with a DDIM-based sampler.

Given that $rank(A) = n$ for $A \in \mathbb{R}^{m \times n}$ with $m \geq n$, we note that $(A^T A)^{-1}$ exists. Consequently, $\lambda_{\min}(A^T A) > 0$. From the theorem statement, we have $\gamma(t) = \frac{2k(T-t+1)}{\lambda_{\min}(A^T A)}$, with $k > 1$. Immediately, we note that for all $t \in [1, T]$, $\gamma(t) > 0$. More specifically, $t \in [1, T]$, $\gamma(t) > \frac{2}{\lambda_{\min}(A^T A)}$.

First, we denote the convergence property for **Algorithm 1** as the variation of the upper bound on the terminal error $\|x^{\mathrm{gen}} - x^*\|_2$, where $x^{\mathrm{gen}}$ is obtained from **Algorithm 1**, as a function of $T$. Particularly, we are interested in how quickly the upper bound on the terminal error decays to reach below a fixed value that is determined by the choice of $k$. Later, we will show that for suitable choices of $k$, the upper bound can decay below the tolerance limit $\delta$.

The proof is divided into 2 parts. First, we obtain the expression for $z_{t-1}$ in terms of $z_t$. Then, we obtain an upper bound for $\|z_0 - x^*\|_2$ or $\|x^{\mathrm{gen}} - x^*\|_2$, as from **Algorithm 1** we note that $z_0 = x^{\mathrm{gen}}$.

First, we note that for deterministic sampling, we have the DDIM control parameters $\sigma_1, \ldots, \sigma_T = 0$. Therefore, the DDIM reverse sampling step from **Algorithm 1** (line 7) can be written as

$$z_{t-1} = \sqrt{\bar{\alpha}_{t-1}}\hat{z}_{0,\mathrm{pr}}(z_t; \epsilon_\theta) + \sqrt{1 - \bar{\alpha}_{t-1}}\epsilon_\theta(z_t, t). \tag{12}$$

Since the true data distribution is Gaussian, the optimal denoiser $\epsilon^*(z_t, t)$ can be expressed analytically for any diffusion step $t$. Therefore, the deterministic sampling step can be written as

$$z_{t-1} = \sqrt{\bar{\alpha}_{t-1}}\hat{z}_{0,\mathrm{pr}}(z_t; \epsilon^*) + \sqrt{1 - \bar{\alpha}_{t-1}}\epsilon^*(z_t, t).$$

We can obtain an analytical expression for the optimal denoiser from **Lemma C.1**. Using Equation 28 from **Lemma C.1**, we note that the optimal denoiser at the diffusion step $t$ is

$$\epsilon^*(z_t, t) = -\sqrt{1 - \bar{\alpha}_t}(\sqrt{\bar{\alpha}_t}\mu - z_t). \tag{13}$$

Now, we obtain the expression for $\hat{z}_{0,\mathrm{pr}}(z_t; \epsilon^*)$. Note that the constraint violation function is defined as $\Pi(z) = \|y - Az\|_2^2$. Consequently, we note that the objective function in line 5 of **Algorithm 1**, i.e., $\frac{1}{2}(\|z - \hat{z}_0(z_t; \epsilon_\theta)\|_2^2 + \gamma(t)\|y - Az\|_2^2)$, is convex with respect to $z$ for $\gamma(t) > 0$. As such, we use **Lemmas C.1** and **C.2** to obtain the expression for $\hat{z}_{0,\mathrm{pr}}(z_t; \epsilon^*)$,

$$\hat{z}_{0,\mathrm{pr}}(z_t; \epsilon^*) = [\mathbf{I}_n + \gamma(t)A^T A]^{-1}[\mu - \bar{\alpha}_t\mu + \sqrt{\bar{\alpha}_t}z_t + \gamma(t)A^T y]. \tag{14}$$

We substitute the expressions for $\epsilon^*(z_t, t)$ from Equation 13 and $\hat{z}_{0,\mathrm{pr}}(z_t; \epsilon^*)$ from Equation 14, respectively, in addition to replacing $y$ with $Ax^*$, to obtain $z_{t-1}$ in terms of $z_t$:

$$\begin{aligned}
z_{t-1} =& \sqrt{\bar{\alpha}_{t-1}} \left[\mathbf{I}_n + \gamma(t)A^T A\right]^{-1} \left[\mu - \bar{\alpha}_t\mu + \sqrt{\bar{\alpha}_t}z_t + \gamma(t)A^T y\right] \\
&+ \sqrt{1 - \bar{\alpha}_{t-1}}(-\sqrt{1 - \bar{\alpha}_t}(\sqrt{\bar{\alpha}_t}\mu - z_t)), \\
z_{t-1} =& \sqrt{\bar{\alpha}_{t-1}} \left[\mathbf{I}_n + \gamma(t)A^T A\right]^{-1} \left[\mu - \bar{\alpha}_t\mu + \sqrt{\bar{\alpha}_t}z_t + \gamma(t)A^T y\right] \\
&- \sqrt{1 - \bar{\alpha}_{t-1}}\sqrt{1 - \bar{\alpha}_t}\sqrt{\bar{\alpha}_t}\mu + \sqrt{1 - \bar{\alpha}_{t-1}}\sqrt{1 - \bar{\alpha}_t}z_t.
\end{aligned}$$

On further simplification, we get

$$\begin{aligned}
z_{t-1} =& \sqrt{\bar{\alpha}_{t-1}} \left[\mathbf{I}_n + \gamma(t)A^T A\right]^{-1} \left[\mu - \bar{\alpha}_t\mu + \sqrt{\bar{\alpha}_t}z_t + \gamma(t)A^T Ax^*\right] \\
&- \sqrt{1 - \bar{\alpha}_{t-1}}\sqrt{1 - \bar{\alpha}_t}\sqrt{\bar{\alpha}_t}\mu + \sqrt{1 - \bar{\alpha}_{t-1}}\sqrt{1 - \bar{\alpha}_t}z_t, \\
z_{t-1} =& \left[\sqrt{\bar{\alpha}_{t-1}}\sqrt{\bar{\alpha}_t} \left[\mathbf{I}_n + \gamma(t)A^T A\right]^{-1} + \sqrt{1 - \bar{\alpha}_{t-1}}\sqrt{1 - \bar{\alpha}_t}\mathbf{I}_n\right] z_t \\
&+ \left[\sqrt{\bar{\alpha}_{t-1}} \left[\mathbf{I}_n + \gamma(t)A^T A\right]^{-1} - \bar{\alpha}_t\sqrt{\bar{\alpha}_{t-1}} \left[\mathbf{I}_n + \gamma(t)A^T A\right]^{-1}\right] \mu \\
&- \left[\sqrt{1 - \bar{\alpha}_{t-1}}\sqrt{1 - \bar{\alpha}_t}\sqrt{\bar{\alpha}_t}\mathbf{I}_n\right] \mu + \gamma(t)\sqrt{\bar{\alpha}_{t-1}} \left[\mathbf{I}_n + \gamma(t)A^T A\right]^{-1} A^T Ax^*, \\
z_{t-1} =& \left[\sqrt{\bar{\alpha}_{t-1}}\sqrt{\bar{\alpha}_t} \left[\mathbf{I}_n + \gamma(t)A^T A\right]^{-1} + \sqrt{1 - \bar{\alpha}_{t-1}}\sqrt{1 - \bar{\alpha}_t}\mathbf{I}_n\right] z_t \\
&+ \left[(1 - \bar{\alpha}_t)\sqrt{\bar{\alpha}_{t-1}} \left[\mathbf{I}_n + \gamma(t)A^T A\right]^{-1}\right] \mu - \left[\sqrt{1 - \bar{\alpha}_{t-1}}\sqrt{1 - \bar{\alpha}_t}\sqrt{\bar{\alpha}_t}\mathbf{I}_n\right] \mu \\
&+ \gamma(t)\sqrt{\bar{\alpha}_{t-1}} \left[\mathbf{I}_n + \gamma(t)A^T A\right]^{-1} A^T Ax^*.
\end{aligned}$$

To this end, we obtain

$$z_{t-1} = K_t z_t + E_t\mu - F_t\mu + \gamma(t)\sqrt{\bar{\alpha}_{t-1}} \left[\mathbf{I}_n + \gamma(t)A^T A\right]^{-1} A^T Ax^*,$$

where we have the following matrix definitions,

$$K_t = \left[\sqrt{\bar{\alpha}_{t-1}}\sqrt{\bar{\alpha}_t} \left[\mathbf{I}_n + \gamma(t)A^T A\right]^{-1} + \sqrt{1 - \bar{\alpha}_{t-1}}\sqrt{1 - \bar{\alpha}_t}\mathbf{I}_n\right], \tag{15}$$

$$E_t = \left[(1 - \bar{\alpha}_t)\sqrt{\bar{\alpha}_{t-1}} \left[\mathbf{I}_n + \gamma(t)A^T A\right]^{-1}\right], \tag{16}$$

$$F_t = \left[\sqrt{1 - \bar{\alpha}_{t-1}}\sqrt{1 - \bar{\alpha}_t}\sqrt{\bar{\alpha}_t}\mathbf{I}_n\right]. \tag{17}$$

The goal is to obtain the upper bound for $\|x^{\mathrm{gen}} - x^*\|_2$. Note that $\|x^{\mathrm{gen}} - x^*\|_2 = \|z_0 - x^*\|_2$. So, first, we subtract $x^*$ from both sides to obtain

$$z_{t-1} - x^* = K_t z_t + E_t\mu - F_t\mu + \gamma(t)\sqrt{\bar{\alpha}_{t-1}} \left[\mathbf{I}_n + \gamma(t)A^T A\right]^{-1} A^T Ax^* - x^*.$$

Further, we add and subtract $K_t x^*$ to the right side to obtain

$$z_{t-1} - x^* = K_t z_t - K_t x^* + E_t\mu - F_t\mu + \gamma(t)\sqrt{\bar{\alpha}_{t-1}} \left[\mathbf{I}_n + \gamma(t)A^T A\right]^{-1} A^T Ax^* - x^* + K_t x^*.$$

We further simplify the above expression to obtain

$$z_{t-1} - x^* = K_t(z_t - x^*) + E_t\mu - F_t\mu + K_t x^* + D_t x^*,$$

where the matrix definition of $D_t$ is

$$D_t = \gamma(t)\sqrt{\bar{\alpha}_{t-1}}\left[\mathbf{I}_n + \gamma(t)A^TA\right]^{-1}A^TA - \mathbf{I}_n. \tag{18}$$

Now, we obtain the expression for $\|z_{t-1} - x^*\|_2$ in terms of $\|z_t - x^*\|_2$.

$$\|z_{t-1} - x^*\|_2 = \|K_t(z_t - x^*) + E_t\mu - F_t\mu + K_t x^* + D_t x^*\|_2.$$

Applying the triangle inequality repeatedly, we get

$$\|z_{t-1} - x^*\|_2 \le \|K_t(z_t - x^*)\|_2 + \|K_t x^*\|_2 + \|D_t x^*\|_2 + \|E_t\mu\|_2 + \|F_t\mu\|_2. \tag{19}$$

Before obtaining the upper bound for $\|z_0 - x^*\|$, for $\gamma(t) > 0$, we will first show that $\|K_t\|_2, \|D_t\|_2, \|E_t\|_2, \|F_t\|_2 < 1 \ \forall \ t \ \in [1, T]$. Here $\|K_t\|_2$ refers to the spectral norm of the matrix $K_t$. To show this, we establish a few relationships that will be the recurring theme used in proving that $\|K_t\|_2, \|D_t\|_2, \|E_t\|_2, \|F_t\|_2 < 1 \ \forall \ t \ \in [1, T]$.

The spectral norm of the matrix M is defined as $\|M\|_2 = \max_{x\neq 0} \frac{\|Mx\|_2}{\|x\|_2}$. From this definition, we immediately note the following two inequalities:

- $\|Mx\|_2 \le \|M\|_2\|x\|_2$ as $\|M\|_2 = \max_{x\neq 0} \frac{\|Mx\|_2}{\|x\|_2}$,
- $\|MN\|_2 = \max_{x\neq 0} \frac{\|MNx\|_2}{\|x\|_2} \le \max_{x\neq 0} \frac{\|M\|_2\|Nx\|_2}{\|x\|_2} \le \max_{x\neq 0} \frac{\|M\|_2\|N\|_2\|x\|_2}{\|x\|_2} = \|M\|_2\|N\|_2$.

Further, we note that the following are well-established properties for spectral norms and positive definite matrices. Consider a positive definite matrix $M$, *i.e.*, $M \succ 0$. Then, we have:

- $\|M\|_2$ is equal to the largest eigen value of $M$, *i.e.*, $\lambda_{\max}(M)$,
- $\|M^{-1}\|_2 = \frac{1}{\lambda_{\min}(M)}$ as the eigenvalues of $M^{-1}$ are the reciprocal of the eigenvalues of $M$,
- $\| - M\|_2 = \|M\|_2$.

We refer the readers to **Lemmas C.3, C.8,** and **C.10**, where we show that $\|K_t\|_2, \|E_t\|_2, \|F_t\|_2 < 1 \ \forall \ t \ \in [1, T]$, if $\gamma(t) > 0$.

Similarly, **Lemma C.6** shows that $\|D_t\|_2 < 1 \ \forall \ t \ \in [1, T]$, if $\gamma(t) > \frac{2}{\lambda_{\min} A^TA}$.

We first apply the inequality $\|Mx\|_2 \le \|M\|_2\|x\|_2$ to simplify Equation 19 as follows:

$$\|z_{t-1} - x^*\|_2 \le \|K_t\|_2\|z_t - x^*\|_2 + \|K_t x^*\|_2 + \|D_t x^*\|_2 + \|E_t\mu\|_2 + \|F_t\mu\|_2. \tag{20}$$

Therefore, we can recursively obtain the upper bound for $\|z_t - x^*\|_2$ in terms of $\|z_T - x^*\|_2$. This process, repeated $T$ times, provides the upper bound for $\|z_0 - x^*\|_2$.

$$\begin{aligned}
\|z_0 - x^*\|_2 \le\ & \|K_1\|_2\|K_2\|_2\ldots\|K_T\|_2\|(z_T - x^*)\|_2 \\
& + (\|K_1\|_2 + \|K_1\|_2\|K_2\|_2 + \cdots + \|K_1\|_2\|K_2\|_2\ldots\|K_{T-1}\|_2\|K_T\|_2)\|x^*\|_2 \\
& + (\|D_1\|_2 + \|K_1\|_2\|D_2\|_2 + \cdots + \|K_1\|_2\|K_2\|_2\ldots\|K_{T-1}\|_2\|D_T\|_2)\|x^*\|_2 \\
& + (\|E_1\|_2 + \|K_1\|_2\|E_2\|_2 + \cdots + \|K_1\|_2\|K_2\|_2\ldots\|K_{T-1}\|_2\|E_T\|_2)\|\mu\|_2 \\
& + (\|F_1\|_2 + \|K_1\|_2\|F_2\|_2 + \cdots + \|K_1\|_2\|K_2\|_2\ldots\|K_{T-1}\|_2\|F_T\|_2)\|\mu\|_2.
\end{aligned} \tag{21}$$

Let $\lambda_k = \max_t(\|K_1\|_2, \|K_2\|_2, \ldots, \|K_T\|_2)$. Since for $\gamma(t) > 0$, $\|K_1\|_2, \ldots, \|K_T\|_2 < 1$, we note that $\lambda_k < 1$.

Therefore, $\|K_1\|_2\|K_2\|_2\ldots\|K_T\|_2$ can be upper bounded by $\lambda_k^T$.

Additionally, note that $\|K_1\|_2\|K_2\|_2 \le \|K_1\|_2$ as $\|K_2\|_2 < 1$. Therefore, $(\|K_1\|_2 + \|K_1\|_2\|K_2\|_2 + \cdots + \|K_1\|_2\|K_2\|_2\ldots\|K_{T-1}\|_2\|K_T\|_2)$ can be upper bounded by $T\|K_1\|_2$.

Similarly, $(\|K_1\|_2\|D_2\|_2 + \cdots + \|K_1\|_2\|K_2\|_2\ldots\|K_{T-1}\|_2\|D_T\|_2)$ can be upperbounded by $(T - 1)\|K_1\|_2$.

The same applies to $(\|K_1\|_2\|E_2\|_2 + \cdots + \|K_1\|_2\|K_2\|_2 \ldots \|K_{T-1}\|_2\|E_T\|_2)$ and $(\|K_1\|_2\|F_2\|_2 + \cdots + \|K_1\|_2\|K_2\|_2 \ldots \|K_{T-1}\|_2\|F_T\|_2)$.

Therefore, the upper bound in Equation 21 can be simplified as

$$\|z_0 - x^*\| \leq \lambda_k^T \|(z_T - x^*)\|_2 + T\|K_1\|_2\|x^*\|_2 + (\|D_1\|_2 + (T-1)\|K_1\|_2)\|x^*\|_2$$
$$+ (\|E_1\|_2 + (T-1)\|K_1\|_2)\|\mu\|_2 + (\|F_1\|_2 + (T-1)\|K_1\|_2)\|\mu\|_2. \quad (22)$$

Consequently, in **Lemmas C.4, C.7, C.9, C.10**, we show

$$\|K_1\|_2 \leq \frac{\sqrt{\bar{\alpha}_1}}{1 + \gamma(1)\lambda_{\min}(A^T A)} < 1 \text{ if } \gamma(1) > 0,$$

$$\|D_1\|_2 \leq \frac{1}{\gamma(1)\lambda_{\min}(A^T A))-1} < 1 \text{ if } \gamma(1) > \frac{2}{\lambda_{\min}(A^T A)},$$

$$\|E_1\|_2 \leq \frac{1 - \bar{\alpha}_1}{1 + \gamma(1)\lambda_{\min}(A^T A)} < 1 \text{ if } \gamma(1) > 0,$$

$$\|F_1\|_2 = 0. \quad (23)$$

For our choice of $\gamma(1) = \frac{2kT}{\lambda_{\min}(A^T A)}$, we first note that $\gamma(1) > 0$ and $\gamma(1) > \frac{2}{\lambda_{\max}(A^T A)}$ for $k > 1$. Therefore, we can rewrite the above inequalities as

$$\|K_1\|_2 \leq \frac{\sqrt{\bar{\alpha}_1}}{1 + 2kT},$$

$$\|D_1\|_2 \leq \frac{1}{2kT-1},$$

$$\|E_1\|_2 \leq \frac{1 - \bar{\alpha}_1}{1 + 2kT},$$

$$\|F_1\|_2 = 0. \quad (24)$$

Therefore, Equation 22 can be upper bounded using Equation 24 as shown below:

$$\|z_0 - x^*\|_2 \leq \lambda_k^T \|(z_T - x^*)\|_2 + T\left(\frac{\sqrt{\bar{\alpha}_1}}{1 + 2kT}\right)\|x^*\|_2 + \left(\frac{1}{2kT-1}\right)\|x^*\|_2 +$$
$$\left(\frac{1 - \bar{\alpha}_1}{1 + 2kT}\right)\|\mu\|_2 + (T-1)\left(\frac{\sqrt{\bar{\alpha}_1}}{1 + 2kT}\right)\|x^*\|_2 + 2(T-1)\left(\frac{\sqrt{\bar{\alpha}_1}}{1 + 2kT}\right)\|\mu\|_2. \quad (25)$$

This can be further simplified to

$$\|z_0 - x^*\|_2 \leq \lambda_k^T \|(z_T - x^*)\|_2 + \left(\frac{\sqrt{\bar{\alpha}_1}}{(1/T) + 2k}\right)\|x^*\|_2 + \left(\frac{1}{2kT-1}\right)\|x^*\|_2 +$$
$$\left(\frac{1 - \bar{\alpha}_1}{1 + 2kT}\right)\|\mu\|_2 + (1 - 1/T)\left(\frac{\sqrt{\bar{\alpha}_1}}{(1/T) + 2k}\right)\|x^*\|_2 +$$
$$2(1 - 1/T)\left(\frac{\sqrt{\bar{\alpha}_1}}{(1/T) + 2k}\right)\|\mu\|_2.$$

Upper bounding $\left(\frac{\sqrt{\bar{\alpha}_1}}{(1/T)+2k}\right)\|x^*\|_2$ by $\left(\frac{\sqrt{\bar{\alpha}_1}}{2k}\right)\|x^*\|_2$, $(1 - 1/T)\left(\frac{\sqrt{\bar{\alpha}_1}}{(1/T)+2k}\right)\|x^*\|_2$ by $\left(\frac{\sqrt{\bar{\alpha}_1}}{2k}\right)\|x^*\|_2$, and $2(1 - 1/T)\left(\frac{\sqrt{\bar{\alpha}_1}}{(1/T)+2k}\right)\|\mu\|_2$ by $\left(\frac{\sqrt{\bar{\alpha}_1}}{2k}\right)\|\mu\|_2$, we get

$$\|z_0 - x^*\|_2 \leq \lambda_k^T \|(z_T - x^*)\|_2 + \left(\frac{1}{2kT-1}\right)\|x^*\|_2 + \left(\frac{1 - \bar{\alpha}_1}{1 + 2kT}\right)\|\mu\|_2 +$$
$$\left(\frac{\sqrt{\bar{\alpha}_1}}{k}\right)(\|\mu\|_2 + \|x^*\|_2). \quad (26)$$

$\left(\frac{\sqrt{\bar{\alpha}_1}}{k}\right)(\|\mu\|_2 + \|x^*\|_2)$ is a constant.

$\lambda_k^T \|(z_T - x^*)\|_2$ converges faster than $\left(\frac{1}{2kT-1}\right)\|x^*\|_2 + \left(\frac{1-\bar{\alpha}_1}{1+2kT}\right)\|\mu\|_2$. Note that $\lambda_k < 1$.

Therefore, **the convergence rate for the upper bound on the terminal error is $\mathcal{O}(1/T)$.**

As $T \to \infty$, we observe the following:

$$\lim_{T\to\infty} \lambda_k^T \|(z_T - x^*)\|_2 = 0 \quad (\lambda_k < 1),$$

$$\lim_{T\to\infty} T\left(\frac{\sqrt{\bar{\alpha}_1}}{1+2kT}\right)\|x^*\|_2 = \left(\frac{\sqrt{\bar{\alpha}_1}}{2k}\right)\|x^*\|_2 \quad (\text{if } k > 0),$$

$$\lim_{T\to\infty} \left(\frac{1}{2kT-1}\right)\|x^*\|_2 = 0,$$

$$\lim_{T\to\infty} \left(\frac{1-\bar{\alpha}_1}{1+2kT}\right)\|\mu\|_2 = 0,$$

$$\lim_{T\to\infty} (T-1)\left(\frac{\sqrt{\bar{\alpha}_1}}{1+2kT}\right)\|x^*\|_2 = \left(\frac{\sqrt{\bar{\alpha}_1}}{2k}\right)\|x^*\|_2 \quad (\text{if } k > 0),$$

$$\lim_{T\to\infty} 2(T-1)\left(\frac{\sqrt{\bar{\alpha}_1}}{1+2kT}\right)\|\mu\|_2 = \left(\frac{\sqrt{\bar{\alpha}_1}}{k}\right)\|\mu\|_2 \quad (\text{if } k > 0).$$

Therefore, in the limit $T \to \infty$, we have

$$\|z_0 - x^*\|_2 \le \frac{\sqrt{\bar{\alpha}_1}}{k}\left(\|x^*\|_2 + \|\mu\|_2\right) \text{ or,}$$

$$\|x^{\text{gen}} - x^*\|_2 \le \frac{\sqrt{\bar{\alpha}_1}}{k}\left(\|x^*\|_2 + \|\mu\|_2\right).$$

Consequently, for the terminal error $\|x^{\text{gen}} - x^*\|_2$ to be upper bounded by $\delta$, which is the desired tolerance limit, the following inequality has to be true:

$$\|x^{\text{gen}} - x^*\|_2 \le \frac{\sqrt{\bar{\alpha}_1}}{k}\left(\|x^*\|_2 + \|\mu\|_2\right) \le \delta. \tag{27}$$

This can be ensured for $k \ge \frac{\sqrt{\bar{\alpha}_1}}{\delta}\left(\|x^*\|_2 + \|\mu\|_2\right)$.

However, from Equation 24, we note that $k$ has to be greater than 1.

Therefore, for $k > \max\left(1, \frac{\sqrt{\bar{\alpha}_1}}{\delta}\left(\|x^*\|_2 + \|\mu\|_2\right)\right)$, we have $\|x^{\text{gen}} - x^*\|_2 \le \delta$.

**Lemma C.1.** *Suppose **Assumption** 4.3 holds. Consider a $T$-step diffusion process with coefficients $\bar{\alpha}_0, \ldots, \bar{\alpha}_T$ such that $\bar{\alpha}_0 = 1$, $\bar{\alpha}_T = 0$, $\bar{\alpha}_t \in [0, 1]$. The optimal denoiser $\epsilon^*(z_t, t)$ is given by*

$$\epsilon^*(z_t, t) = -\sqrt{1 - \bar{\alpha}_t}(\sqrt{\bar{\alpha}_t}\mu - z_t).$$

*Proof.* We first observe the distribution of $z_t$. For the diffusion forward process, we know that $z_t = \sqrt{\bar{\alpha}_t}z_0 + \sqrt{1 - \bar{\alpha}_t}\epsilon$, where $\epsilon \sim \mathcal{N}(\mathbf{0}_n, \mathbf{I}_n)$. Note that $z_0$ is a sample from the Gaussian distribution $\mathcal{N}(\mu, \mathbf{I}_n)$. Consequently, we note that $z_t$ is a sample from the Gaussian distribution $\mathcal{N}(\sqrt{\bar{\alpha}_t}\mu + 0_n, \bar{\alpha}_t\mathbf{I}_n + (1-\bar{\alpha}_t)\mathbf{I}_n)$. On simplification, we note that $z_t$ is a sample from $\mathcal{N}(\sqrt{\bar{\alpha}_t}\mu, \mathbf{I}_n)$. We denote the PDF of $z_t$'s marginal distribution as $q_t(z_t)$. Since we are using the optimal denoiser, the reverse process PDF at $t$, induced by the optimal denoiser, $p_{*,t}(z_t)$ is the same as the forward process PDF at $t$, which is $q_t(z_t)$. Here, note that in Section 2.1, we denote the reverse process PDF as $p_{\theta,t}$, where the reverse process is governed by the denoiser $\epsilon_\theta$. We replace this notation with $p_{*,t}(z_t)$ as we are using the optimal denoiser.

Therefore, the score function at $t$ is given by $\nabla_{z_t}\log p_{*,t}(z_t) = \nabla_{z_t}\log q_t(z_t)$. The score function for the Gaussian distribution $q_t(z_t)$ with mean $\sqrt{\bar{\alpha}_t}\mu$ and covariance matrix $\mathbf{I}_n$, *i.e.*, $\nabla_{z_t}(\log q_t(z_t))$ is given by $\sqrt{\bar{\alpha}_t}\mu - z_t$. Finally, [55] shows that for the diffusion step $t$, the optimal denoiser can be obtained from the score function using the following expression:

$$\epsilon^*(z_t, t) = -\sqrt{1 - \bar{\alpha}_t}\nabla_{z_t}\log q_t(z_t) \Rightarrow \epsilon^*(z_t, t) = -\sqrt{1 - \bar{\alpha}_t}(\sqrt{\bar{\alpha}_t}\mu - z_t). \tag{28}$$

$\square$

**Lemma C.2.** *Suppose Assumption 4.3 holds. Consider a $T$-step diffusion process with coefficients $\bar{\alpha}_0, \ldots, \bar{\alpha}_T$ such that $\bar{\alpha}_0 = 1$, $\bar{\alpha}_T = 0$, $\bar{\alpha}_t \in [0,1]$. The projected posterior mean estimate, $\hat{z}_{0,\mathrm{pr}}(z_t; \epsilon_\theta)$, from the projection step in line 5 of **Algorithm 1** is given by*

$$\hat{z}_{0,\mathrm{pr}}(z_t; \epsilon_\theta) = [I + \gamma(t)A^T A]^{-1}[\mu - \bar{\alpha}_t\mu + \sqrt{\bar{\alpha}_t}z_t + \gamma(t)A^T y],$$

*where the penalty coefficients from **Algorithm 1**, $\gamma(t)$, are non-negative, i.e., $\gamma(t) > 0 \ \forall \ t \in [1, \ldots, T]$.*

*Proof.* We start with the unconstrained minimization in line 5 of **Algorithm 1**, given by

$$\hat{z}_{0,\mathrm{pr}}(z_t; \epsilon_\theta) = \arg\min_z \frac{1}{2} \left( \|z - \hat{z}_0(z_t; \epsilon_\theta)\|_2^2 + \gamma(t)\|y - Az\|_2^2 \right).$$

Note that we replaced the penalty function $\Pi(z)$ with $\|y - Az\|_2^2$, as we are required to generate a sample that satisfies the constraint $y = Az$.

Since the objective function is convex with respect to $z$, we obtain the global minimum by setting the gradient with respect to $z$ to 0, *i.e.,*

$$\nabla_z \left( \frac{1}{2} \left( \|z - \hat{z}_0(z_t; \epsilon_\theta)\|_2^2 + \gamma(t)\|y - Az\|_2^2 \right) \right) = 0,$$

$$\nabla_z \left( \frac{1}{2} \left( z^T z - 2z^T \hat{z}_0(z_t; \epsilon_\theta) + \hat{z}_0(z_t; \epsilon_\theta)^T \hat{z}_0(z_t; \epsilon_\theta) \right) \right) + \gamma(t)\nabla_z \left( \frac{1}{2}\|y - Az\|_2^2 \right) = 0,$$

$$z - \hat{z}_0(z_t; \epsilon_\theta) + \gamma(t)\nabla_z \left( \frac{1}{2}\|y - Az\|_2^2 \right) = 0,$$

$$z - \hat{z}_0(z_t; \epsilon_\theta) + \gamma(t)\nabla_z \left( \frac{1}{2} \left( y^T y + z^T A^T Az - 2y^T Az \right) \right) = 0,$$

$$z - \hat{z}_0(z_t; \epsilon_\theta) + \gamma(t) \left( A^T Az - A^T y \right) = 0,$$

$$\left[ \mathbf{I}_n + \gamma(t)A^T A \right] z - \left( \hat{z}_0(z_t; \epsilon_\theta) + \gamma(t)A^T y \right) = 0.$$

Solving this, we obtain the following expression for $\hat{z}_{0,\mathrm{pr}}(z_t; \epsilon_\theta)$:

$$\hat{z}_{0,\mathrm{pr}}(z_t; \epsilon_\theta) = [\mathbf{I}_n + \gamma(t)A^T A]^{-1}(\hat{z}_0(z_t; \epsilon_\theta) + \gamma(t)A^T y).$$

Note that the inverse of $\left[ \mathbf{I}_n + \gamma(t)A^T A \right]$ exists as $A^T A \succ 0$ (from **Assumption 4.3**) and $\gamma(t) > 0$, which ensures $\left[ \mathbf{I}_n + \gamma(t)A^T A \right] \succ 0$. Further, substituting the expression for $\hat{z}_0(z_t; \epsilon_\theta)$, we obtain

$$\hat{z}_{0,\mathrm{pr}}(z_t; \epsilon_\theta) = [\mathbf{I}_n + \gamma(t)A^T A]^{-1} \left[ \frac{z_t - \sqrt{1 - \bar{\alpha}_t}\epsilon_\theta(z_t, t)}{\sqrt{\bar{\alpha}_t}} + \gamma(t)A^T y \right].$$

Given that $P_{\mathrm{data}} = \mathcal{N}(\mu, \mathbf{I}_n)$, for the $T$-step diffusion process with coefficients $\bar{\alpha}_0, \ldots, \bar{\alpha}_T$, we use the expression for the optimal denoiser $\epsilon^*(z_t, t)$ (check Equation 28) in place of $\epsilon_\theta(z_t, t)$ to obtain

$$\hat{z}_{0,\mathrm{pr}}(z_t; \epsilon^*) = [\mathbf{I}_n + \gamma(t)A^T A]^{-1} \left[ \frac{z_t + (1 - \bar{\alpha}_t)(\sqrt{\bar{\alpha}_t}\mu - z_t)}{\sqrt{\bar{\alpha}_t}} + \gamma(t)A^T y \right],$$

$$\hat{z}_{0,\mathrm{pr}}(z_t; \epsilon^*) = [\mathbf{I}_n + \gamma(t)A^T A]^{-1} \left[ \frac{z_t + \sqrt{\bar{\alpha}_t}\mu - z_t - \bar{\alpha}_t\sqrt{\bar{\alpha}_t}\mu + \bar{\alpha}_t z_t}{\sqrt{\bar{\alpha}_t}} + \gamma(t)A^T y \right].$$

This can be finally simplified to obtain the expression

$$\hat{z}_{0,\mathrm{pr}}(z_t; \epsilon^*) = [\mathbf{I}_n + \gamma(t)A^T A]^{-1}[\mu - \bar{\alpha}_t\mu + \sqrt{\bar{\alpha}_t}z_t + \gamma(t)A^T y].$$

$\square$

**Lemma C.3.** *Suppose Assumption 4.3 holds. Consider a $T$-step diffusion process with coefficients $\bar{\alpha}_0, \ldots, \bar{\alpha}_T$ such that $\bar{\alpha}_0 = 1$, $\bar{\alpha}_T = 0$, $\bar{\alpha}_t \in [0,1]$. If $\bar{\alpha}_t < \bar{\alpha}_{t-1}$ and if the penalty coefficients from **Algorithm 1** are given by $\gamma(t) > 0 \ \forall \ t \in [1, T]$, the spectral norm of the matrix $K_t$, $\|K_t\|_2$, with $K_t$ defined as in Equation 15, is less than 1.*

*Proof.* We want to show that

$$\|K_t\|_2 = \left\| \left[ \sqrt{\bar{\alpha}_{t-1}}\sqrt{\bar{\alpha}_t} \left[ \mathbf{I}_n + \gamma(t)A^T A \right]^{-1} + \sqrt{1-\bar{\alpha}_{t-1}}\sqrt{1-\bar{\alpha}_t}\mathbf{I}_n \right] \right\|_2 < 1.$$

The spectral norm follows the triangle inequality. Therefore, after simplifying the expression with triangle inequality, we need to show

$$\left\| \sqrt{\bar{\alpha}_{t-1}}\sqrt{\bar{\alpha}_t} \left[ \mathbf{I}_n + \gamma(t)A^T A \right]^{-1} \right\|_2 + \left\| \sqrt{1-\bar{\alpha}_{t-1}}\sqrt{1-\bar{\alpha}_t}\mathbf{I}_n \right\|_2 < 1.$$

We note that for $\gamma(t) > 0$, $\left[ \mathbf{I}_n + \gamma(t)A^T A \right] \succ 0$, and therefore $\left[ \mathbf{I}_n + \gamma(t)A^T A \right]^{-1} \succ 0$. Similarly, $\mathbf{I}_n \succ 0$.

Further, we use the identities that if $M \succ 0$, then $\|M\|_2 = \lambda_{\max}(M)$, $\|M^{-1}\|_2 = \frac{1}{\lambda_{\min}(M)}$, and $\|cM\|_2 = |c|\|M\|_2$.

Therefore, $\|\mathbf{I}_n\|_2 = 1$, $\| \left[ \mathbf{I}_n + \gamma(t)A^T A \right]^{-1} \|_2 = \frac{1}{\lambda_{\min}([\mathbf{I}_n + \gamma(t)A^T A])}$. Further, note that $\sqrt{\bar{\alpha}_{t-1}}\sqrt{\bar{\alpha}_t} \geq 0$ and $\sqrt{1-\bar{\alpha}_{t-1}}\sqrt{1-\bar{\alpha}_t} \geq 0$. Substituting these, the inequality simplifies to

$$\frac{\sqrt{\bar{\alpha}_{t-1}}\sqrt{\bar{\alpha}_t}}{\lambda_{\min}([\mathbf{I}_n + \gamma(t)A^T A])} + \sqrt{1-\bar{\alpha}_{t-1}}\sqrt{1-\bar{\alpha}_t} < 1.$$

Therefore, it is sufficient to show that

$$\frac{\sqrt{\bar{\alpha}_{t-1}}\sqrt{\bar{\alpha}_t}}{\lambda_{\min}([\mathbf{I}_n + \gamma(t)A^T A])} < 1 - \sqrt{1-\bar{\alpha}_{t-1}}\sqrt{1-\bar{\alpha}_t}.$$

For any diffusion process with noise coefficients $\bar{\alpha}_0, \ldots, \bar{\alpha}_T$, where $\bar{\alpha}_t > \bar{\alpha}_{t-1} \ \forall \ t \in [1, T]$, **Lemma C.5** shows that $\sqrt{\bar{\alpha}_{t-1}}\sqrt{\bar{\alpha}_t} \leq 1 - \sqrt{1-\bar{\alpha}_{t-1}}\sqrt{1-\bar{\alpha}_t}$. Therefore, it is sufficient to show that $\lambda_{\min}\left( \left[ \mathbf{I}_n + \gamma(t)A^T A \right] \right) > 1$.

To proceed further, we use the Weyl's inequality [56], which states that for any two real symmetric matrices $P \in \mathbb{R}^{n \times n}$ and $Q \in \mathbb{R}^{n \times n}$, if the eigenvalues are represented as $\lambda_{\max}(P) = \lambda_1(P) >= \lambda_2(P) \cdots >= \lambda_n(P) = \lambda_{\min}(P)$, and $\lambda_{\max}(Q) = \lambda_1(Q) >= \lambda_2(Q) \cdots >= \lambda_n(Q) = \lambda_{\min}(Q)$, then we have the following inequality:

$$\lambda_i(P) + \lambda_j(Q) \leq \lambda_{i+j-n}(P + Q). \tag{29}$$

For $i = j = n$, we have $\lambda_{\min}(P) + \lambda_{\min}(Q) \leq \lambda_{\min}(P + Q)$.

For $P = \mathbf{I}_n$ and $Q = \gamma(t)A^T A$ with $\gamma(t) > 0$, this inequality can be exploited as both these matrices are real and symmetric. Therefore, we have

$$\lambda_{\min}\left( \left[ \mathbf{I}_n + \gamma(t)A^T A \right] \right) \geq \lambda_{\min}(\mathbf{I}_n) + \lambda_{\min}(\gamma(t)A^T A), \tag{30}$$

$$\lambda_{\min}\left( \left[ \mathbf{I}_n + \gamma(t)A^T A \right] \right) \geq 1 + \gamma(t)\lambda_{\min}(A^T A). \tag{31}$$

Note that now it is sufficient to show $1 + \gamma(t)\lambda_{\min}(A^T A) > 1$. For $\gamma(t) > 0$, this inequality holds true as $\lambda_{\min}(A^T A) > 0$ $(A^T A \succ 0)$. Therefore,

$$\left\| \left[ \sqrt{\bar{\alpha}_{t-1}}\sqrt{\bar{\alpha}_t} \left[ \mathbf{I}_n + \gamma(t)A^T A \right]^{-1} + \sqrt{1-\bar{\alpha}_{t-1}}\sqrt{1-\bar{\alpha}_t}\mathbf{I}_n \right] \right\|_2 < 1.$$

$\square$

**Lemma C.4.** *Suppose **Assumption 4.3** holds. Consider a $T$-step diffusion process with coefficients $\bar{\alpha}_0, \ldots, \bar{\alpha}_T$ such that $\bar{\alpha}_0 = 1$, $\bar{\alpha}_T = 0$, $\bar{\alpha}_t \in [0, 1]$. If $\bar{\alpha}_t < \bar{\alpha}_{t-1}$ and the penalty coefficients from **Algorithm 1** given by $\gamma(t) > 0 \ \forall \ t \in [1, T]$, $\|K_1\|_2$ with $K_t$ defined as in Equation 15 is given by*

$$\|K_1\|_2 \leq \frac{\sqrt{\bar{\alpha}_1}}{1 + \gamma(1)\lambda_{\min}(A^T A)}. \tag{32}$$

*Proof.* We want to find an upper bound for

$$\|K_t\|_2 = \left\| \left[ \sqrt{\bar{\alpha}_{t-1}}\sqrt{\bar{\alpha}_t} \left[ \mathbf{I}_n + \gamma(t)A^T A \right]^{-1} + \sqrt{1 - \bar{\alpha}_{t-1}}\sqrt{1 - \bar{\alpha}_t}\mathbf{I}_n \right] \right\|_2.$$

Applying the triangle inequality for spectral norm, we get

$$\|K_t\|_2 \leq \left\| \sqrt{\bar{\alpha}_{t-1}}\sqrt{\bar{\alpha}_t} \left[ \mathbf{I}_n + \gamma(t)A^T A \right]^{-1} \right\|_2 + \left\| \sqrt{1 - \bar{\alpha}_{t-1}}\sqrt{1 - \bar{\alpha}_t}\mathbf{I}_n \right\|_2.$$

We use the same simplifications shown in **Lemma C.3** to obtain

$$\|K_t\|_2 \leq \frac{\sqrt{\bar{\alpha}_{t-1}}\sqrt{\bar{\alpha}_t}}{\lambda_{\min}([\mathbf{I}_n + \gamma(t)A^T A])} + \sqrt{1 - \bar{\alpha}_{t-1}}\sqrt{1 - \bar{\alpha}_t}.$$

For $t = 1$, we know that $\bar{\alpha}_{t-1} = \bar{\alpha}_0 = 1$. Therefore, we obtain

$$\|K_1\|_2 \leq \frac{\sqrt{\bar{\alpha}_1}}{\lambda_{\min}([\mathbf{I}_n + \gamma(1)A^T A])}.$$

Further, the denominator can be lower bounded using Weyl's inequality, as shown in Equation 31. Therefore, we obtain

$$\|K_1\|_2 \leq \frac{\sqrt{\bar{\alpha}_1}}{\lambda_{\min}([\mathbf{I}_n + \gamma(1)A^T A])} \leq \frac{\sqrt{\bar{\alpha}_1}}{1 + \gamma(1)\lambda_{\min}(A^T A)}.$$

Hence, we have shown that

$$\|K_1\|_2 \leq \frac{\sqrt{\bar{\alpha}_1}}{1 + \gamma(1)\lambda_{\min}(A^T A)}.$$

$\square$

**Lemma C.5.** *For any $T$-step diffusion process with coefficients $\bar{\alpha}_0, \ldots, \bar{\alpha}_T$ such that $\bar{\alpha}_0 = 1$, $\bar{\alpha}_T = 0$, $\bar{\alpha}_t \in [0, 1] \ \forall \ t \in [1, T]$, if $\bar{\alpha}_t < \bar{\alpha}_{t-1}$, then*

$$\sqrt{\bar{\alpha}_{t-1}}\sqrt{\bar{\alpha}_t} < 1 - \sqrt{1 - \bar{\alpha}_{t-1}}\sqrt{1 - \bar{\alpha}_t}.$$

*Proof.* Squaring on both sides, we get

$$\bar{\alpha}_{t-1}\bar{\alpha}_t < 1 + (1 - \bar{\alpha}_{t-1})(1 - \bar{\alpha}_t) - 2\sqrt{1 - \bar{\alpha}_{t-1}}\sqrt{1 - \bar{\alpha}_t}.$$

After further simplification, we have to show

$$\bar{\alpha}_{t-1}\bar{\alpha}_t < (1 - \bar{\alpha}_t) + (1 - \bar{\alpha}_{t-1}) + \bar{\alpha}_{t-1}\bar{\alpha}_t - 2\sqrt{1 - \bar{\alpha}_{t-1}}\sqrt{1 - \bar{\alpha}_t},$$
$$0 < (1 - \bar{\alpha}_t) + (1 - \bar{\alpha}_{t-1}) - 2\sqrt{1 - \bar{\alpha}_{t-1}}\sqrt{1 - \bar{\alpha}_t},$$
$$0 < (\sqrt{1 - \bar{\alpha}_{t-1}} - \sqrt{1 - \bar{\alpha}_t})^2.$$

Since $\bar{\alpha}_t \neq \bar{\alpha}_{t-1}$, we know that $\sqrt{1 - \bar{\alpha}_{t-1}} \neq \sqrt{1 - \bar{\alpha}_t}$. Therefore $(\sqrt{1 - \bar{\alpha}_{t-1}} - \sqrt{1 - \bar{\alpha}_t})^2 > 0$. Therefore, we conclude that

$$\sqrt{\bar{\alpha}_{t-1}}\sqrt{\bar{\alpha}_t} < 1 - \sqrt{1 - \bar{\alpha}_{t-1}}\sqrt{1 - \bar{\alpha}_t}.$$

Note that this clearly holds for the edge case $t = 1$, where we have $\sqrt{\bar{\alpha}_1} < 1$, and for $t = T$, where we have $0 < 1 - \sqrt{1 - \bar{\alpha}_{T-1}}$. For the choices of $\bar{\alpha}_0, \ldots, \bar{\alpha}_T$, these clearly hold true. $\square$

**Lemma C.6.** *Suppose Assumption 4.3 holds. Consider a $T$-step diffusion process with coefficients $\bar{\alpha}_0, \ldots, \bar{\alpha}_T$ such that $\bar{\alpha}_0 = 1$, $\bar{\alpha}_T = 0$, $\bar{\alpha}_t \in [0, 1] \ \forall \ t \in [1, T]$. For the penalty coefficients from Algorithm 1 given by $\gamma(t) > \frac{2}{\lambda_{\min}(A^T A)}$, $\|D_t\|_2$, with $D_t$ defined as in Equation 18, is less than 1.*

*Proof.* Note that the matrix $D_t$ is given by,

$$D_t = \gamma(t)\sqrt{\bar{\alpha}_{t-1}} \left[ \mathbf{I}_n + \gamma(t)A^T A \right]^{-1} A^T A - \mathbf{I}_n.$$

Using the matrix inversion identity, $(AB)^{-1} = B^{-1}A^{-1}$, we rewrite $D_t$ as follows.

$$D_t = \gamma(t)\sqrt{\bar{\alpha}_{t-1}} \left[ \left( A^T A \right)^{-1} \left[ \mathbf{I}_n + \gamma(t) A^T A \right] \right]^{-1} - \mathbf{I}_n.$$

$$D_t = \sqrt{\bar{\alpha}_{t-1}} \left[ \frac{\left( A^T A \right)^{-1}}{\gamma(t)} \left[ \mathbf{I}_n + \gamma(t) A^T A \right] \right]^{-1} - \mathbf{I}_n.$$

$$D_t = \sqrt{\bar{\alpha}_{t-1}} \left[ \frac{\left( A^T A \right)^{-1}}{\gamma(t)} + \mathbf{I}_n \right]^{-1} - \mathbf{I}_n.$$

We observe that the choice of $\gamma(t)$ is greater than 0. More precisely, $\gamma(t) > \frac{2}{\lambda_{\min}(A^T A)}$. Now, if $\| -\frac{(A^T A)^{-1}}{\gamma(t)} \|_2 < 1$, then we can apply the Neumann's series for matrix inversion, which states that if $\|M\|_2 < 1$, then

$$[\mathbf{I}_n - M]^{-1} = \sum_{i=0}^{\infty} M^i. \tag{33}$$

First, note that $\| - A \|_2 = \|A\|_2$. Therefore, $\| - \frac{(A^T A)^{-1}}{\gamma(t)} \|_2 = \| \frac{(A^T A)^{-1}}{\gamma(t)} \|_2$ for $\gamma(t) > 0$. From the theorem statement, $\gamma(t) > 0$.

Additionally, we know that $\| \frac{(A^T A)^{-1}}{\gamma(t)} \|_2 = \lambda_{\max} \left( \frac{(A^T A)^{-1}}{\gamma(t)} \right) = \frac{1}{\gamma(t)\lambda_{\min}(A^T A)}$.

Therefore, it is enough to show that $\frac{1}{\gamma(t)\lambda_{\min}(A^T A)} < 1$ to apply the Neumann's series.

However, we know that $\gamma(t) > \frac{2}{\lambda_{\min}((A^T A)^{-1})}$. Therefore, we observe that $\frac{1}{\gamma(t)\lambda_{\min}(A^T A)} < \frac{1}{2} < 1$.

Thus, we have shown that $\| \frac{(A^T A)^{-1}}{\gamma(t)} \|_2 < 1$. Therefore, using Equation 33, we get

$$\left[ \mathbf{I}_n - \left( -\frac{(A^T A)^{-1}}{\gamma(t)} \right) \right]^{-1} = \sum_{i=0}^{\infty} \left( \frac{(-A^T A)^{-1}}{\gamma(t)} \right)^i = \mathbf{I}_n + \sum_{i=1}^{\infty} \left( \frac{(-1)^i (A^T A)^{-i}}{\gamma(t)^i} \right). \tag{34}$$

The last equality stems from the fact that for any matrix $M \in \mathbb{R}^{n \times n}$, $M^0 = \mathbf{I}_n$. Substituting this expression for the second term in $D_t$, we get

$$D_t = \sqrt{\bar{\alpha}_{t-1}} \left( \sum_{i=1}^{\infty} \left( \frac{(-1)^i (A^T A)^{-i}}{\gamma(t)^i} \right) \right) + \sqrt{\bar{\alpha}_{t-1}} \mathbf{I}_n - \mathbf{I}_n.$$

On further simplification, we have

$$D_t = \sqrt{\bar{\alpha}_{t-1}} \left( \sum_{i=1}^{\infty} \left( \frac{(-1)^i (A^T A)^{-i}}{\gamma(t)^i} \right) \right) - \left( 1 - \sqrt{\bar{\alpha}_{t-1}} \right) \mathbf{I}_n.$$

Computing the spectral norm and using the triangle inequality, we get

$$\|D_t\|_2 = \left\| \sqrt{\bar{\alpha}_{t-1}} \left( \sum_{i=1}^{\infty} \left( \frac{(-1)^i (A^T A)^{-i}}{\gamma(t)^i} \right) \right) - \left( 1 - \sqrt{\bar{\alpha}_{t-1}} \right) \mathbf{I}_n \right\|_2,$$

$$\leq \sqrt{\bar{\alpha}_{t-1}} \left( \sum_{i=1}^{\infty} \left\| \frac{(-1)^i (A^T A)^{-i}}{\gamma(t)^i} \right\|_2 \right) + \left\| \left( 1 - \sqrt{\bar{\alpha}_{t-1}} \right) \mathbf{I}_n \right\|_2.$$

The inequality arises from the triangle inequality for spectral norms. Note that each of the matrices within the summation is either positive definite or negative definite, and the spectral norms of all these matrices can be represented as $\left\| \frac{(A^T A)^{-i}}{\gamma(t)^i} \right\|_2$. Therefore, we get

$$\|D_t\|_2 \leq \sqrt{\bar{\alpha}_{t-1}} \left( \sum_{i=1}^{\infty} \left\| \frac{(A^T A)^{-i}}{\gamma(t)^i} \right\|_2 \right) + \left( 1 - \sqrt{\bar{\alpha}_{t-1}} \right).$$

Using the inequality $\|MN\|_2 \le \|M\|_2\|N\|_2$ multiple times, we get the following:

$$\left\|\frac{(A^T A)^{-i}}{\gamma(t)^i}\right\|_2 \le \frac{1}{\gamma(t)^i}\left(\left\|(A^T A)^{-1}\right\|_2\right)^i.$$

Additionally, for the above equation, we used $\|cM\|_2 = |c|\|M\|_2$. Here, $c$ is $\gamma(t)$, which is greater than 0. Since $A^T A \succ 0$, we have $\left\|(A^T A)^{-1}\right\|_2 = \frac{1}{\lambda_{\min}(A^T A)}$. Therefore, we have the following inequality:

$$\left\|\frac{(A^T A)^{-i}}{\gamma(t)^i}\right\|_2 \le \frac{1}{\gamma(t)^i}\left(\frac{1}{\lambda_{\min}(A^T A)}\right)^i = \frac{1}{(\gamma(t)\lambda_{\min}(A^T A))^i}.$$

Using this to upper bound $\|D_t\|_2$, we get

$$\|D_t\|_2 \le \sqrt{\bar{\alpha}_{t-1}}\left(\sum_{i=1}^{\infty}\left(\frac{1}{(\gamma(t)\lambda_{\min}(A^T A))^i}\right)\right) + (1 - \sqrt{\bar{\alpha}_{t-1}}).$$

Finally, the summation of an infinite geometric series of the form $a + a^2 + \ldots$, where $a < 1$ is $\frac{a}{1-a}$. Here, note that we have $\gamma(t) > \frac{1}{\lambda_{\min}(A^T A)}$. Therefore, $\frac{1}{\gamma(t)\lambda_{\min}(A^T A)} < 1$. Therefore, we have,

$$\sum_{i=1}^{\infty}\left(\frac{1}{\gamma(t)^i(\lambda_{\min}(A^T A))^i}\right) = \frac{\frac{1}{\gamma(t)\lambda_{\min}(A^T A)}}{1 - \frac{1}{\gamma(t)\lambda_{\min}(A^T A)}} = \frac{1}{\gamma(t)\lambda_{\min}(A^T A) - 1}.$$

So, we obtain

$$\|D_t\|_2 \le \frac{\sqrt{\bar{\alpha}_{t-1}}}{\gamma(t)\lambda_{\min}(A^T A) - 1} + (1 - \sqrt{\bar{\alpha}_{t-1}}). \tag{35}$$

Now, for $\|D_t\|_2 < 1$, we need to show

$$\frac{\sqrt{\bar{\alpha}_{t-1}}}{\gamma(t)\lambda_{\min}(A^T A) - 1} + (1 - \sqrt{\bar{\alpha}_{t-1}}) < 1, \text{ or}$$

$$\frac{\sqrt{\bar{\alpha}_{t-1}}}{\gamma(t)\lambda_{\min}(A^T A) - 1} < \sqrt{\bar{\alpha}_{t-1}}.$$

This simplifies to showing $\gamma(t)\lambda_{\min}(A^T A) - 1 > 1$, which is true if $\gamma(t) > \frac{2}{\lambda_{\min}(A^T A)}$. And, from the statement of the lemma, we know that $\gamma(t) > \frac{2}{\lambda_{\min}(A^T A)}$.

Therefore, we have shown that $\|D_t\|_2 < 1$ for $\gamma(t) > \frac{2}{\lambda_{\min}(A^T A)}$. $\square$

**Lemma C.7.** *Suppose **Assumption** 4.3 holds. Consider a $T$-step diffusion process with coefficients $\bar{\alpha}_0, \ldots, \bar{\alpha}_T$ such that $\bar{\alpha}_0 = 1$, $\bar{\alpha}_T = 0$, $\bar{\alpha}_t \in [0, 1]$ $\forall\, t \in [0, T]$. For the penalty coefficients from **Algorithm** 1 given by $\gamma(1) > \frac{2}{\lambda_{\min}(A^T A)}$, $\|D_1\|_2$, with $D_t$ defined as in Equation 18, is upper bounded by*

$$\|D_1\|_2 \le \frac{1}{\gamma(1)\lambda_{\min}(A^T A) - 1}.$$

*Proof.* Note that the matrix $D_t$ is given by,

$$D_t = \gamma(t)\sqrt{\bar{\alpha}_{t-1}}\left[\mathbf{I}_n + \gamma(t)A^T A\right]^{-1} A^T A - \mathbf{I}_n.$$

From Equation 35 in **Lemma C.6**, we know that if $\gamma(t) > \frac{1}{\lambda_{\min}(A^T A)}$,

$$\|D_t\|_2 \le \frac{\sqrt{\bar{\alpha}_{t-1}}}{\gamma(t)\lambda_{\min}(A^T A) - 1} + (1 - \sqrt{\bar{\alpha}_{t-1}}).$$

From the lemma, we know that $\gamma(t) > \frac{2}{\lambda_{\min}(A^T A)}$. Therefore, we use Equation 35 and substitute for $t = 1$ and $\bar{\alpha}_0 = 1$, we get

$$\|D_1\|_2 \le \frac{1}{\gamma(1)\lambda_{\min}(A^T A) - 1}.$$

$\square$

**Lemma C.8.** *Suppose **Assumption 4.3** holds. Consider a T-step diffusion process with coefficients $\bar{\alpha}_0, \ldots, \bar{\alpha}_T$ such that $\bar{\alpha}_0 = 1$, $\bar{\alpha}_T = 0$, $\bar{\alpha}_t \in [0, 1]$. If $\bar{\alpha}_t < \bar{\alpha}_{t-1} \forall\ t \in [1, T]$ with the penalty coefficients from **Algorithm 1** given by $\gamma(t) > 0$, $\|E_t\|_2 < 1$ where $E_t$ is defined as in Equation 16.*

*Proof.* We know that the matrix $E_t$ is defined as

$$E_t = \left[ (1 - \bar{\alpha}_t)\sqrt{\bar{\alpha}_{t-1}} \left[ \mathbf{I}_n + \gamma(t)A^T A \right]^{-1} \right].$$

First, we use the identity $\|cM\|_2 = |c| \|M\|_2$, where $c$ is any real number, we need to show

$$(1 - \bar{\alpha}_t)\sqrt{\bar{\alpha}_{t-1}} \left\| \left[ \mathbf{I}_n + \gamma(t)A^T A \right]^{-1} \right\|_2 < 1.$$

Note that $(1 - \bar{\alpha}_t)\sqrt{\bar{\alpha}_{t-1}} \geq 0$. Further, for $\gamma(t) > 0$, $\left[ \mathbf{I}_n + \gamma(t)A^T A \right] \succ 0$, and therefore $\left[ \mathbf{I}_n + \gamma(t)A^T A \right]^{-1} \succ 0$.

We use the identity that for $M \succ 0$, $\|M^{-1}\|_2 = \frac{1}{\lambda_{\min}(M)}$.

Therefore, $\| \left[ \mathbf{I}_n + \gamma(t)A^T A \right]^{-1} \|_2 = \frac{1}{\lambda_{\min}([\mathbf{I}_n + \gamma(t)A^T A])}$. We use this expression to simplify the inequality as

$$\frac{(1 - \bar{\alpha}_t)\sqrt{\bar{\alpha}_{t-1}}}{\lambda_{\min}([\mathbf{I}_n + \gamma(t)A^T A])} < 1.$$

We use Weyl's inequality (check Equation 31) to lower bound the denominator and thereby upper bound the left side. Therefore, it is sufficient to show

$$\frac{(1 - \bar{\alpha}_t)\sqrt{\bar{\alpha}_{t-1}}}{1 + \gamma(t)\lambda_{\min}(A^T A)} < 1.$$

We observe that the numerator $(1 - \bar{\alpha}_t)\sqrt{\bar{\alpha}_{t-1}}$ is always less than 1. However, we know that the denominator $1 + \gamma(t)\lambda_{\min}(A^T A)$ is strictly greater than 1 for $\gamma(t) > 0$ as $(A^T A)^{-1}$ exists and $\lambda_{\min}(A^T A) > 0$. Therefore, the left side is always less than 1. This leads to

$$\left\| (1 - \bar{\alpha}_t)\sqrt{\bar{\alpha}_{t-1}} \left[ \mathbf{I}_n + \gamma(t)A^T A \right]^{-1} \right\|_2 < 1.$$

$\square$

**Lemma C.9.** *Suppose **Assumption 4.3** holds. Consider a T-step diffusion process with coefficients $\bar{\alpha}_0, \ldots, \bar{\alpha}_T$ such that $\bar{\alpha}_0 = 1$, $\bar{\alpha}_T = 0$, $\bar{\alpha}_t \in [0, 1]$, If $\bar{\alpha}_t < \bar{\alpha}_{t-1} \forall\ t \in [1, T]$ with the penalty coefficients from **Algorithm 1** given by $\gamma(t) > 0$, $\|E_1\|_2$, with $E_t$ defined as in Equation 16, is upper bounded by*

$$\sigma_{\max}(E_1) \leq \frac{1 - \bar{\alpha}_1}{1 + \gamma(1)\lambda_{\min}(A^T A)}.$$

*Proof.* We know that $E_t$ is given by

$$E_t = \sqrt{\bar{\alpha}_{t-1}}(1 - \bar{\alpha}_t) \left[ \mathbf{I}_n + \gamma(t)A^T A \right]^{-1}.$$

We first substitute for $t = 1$ and $\sqrt{\bar{\alpha}_0} = 1$

$$E_1 = (1 - \bar{\alpha}_1)[\mathbf{I}_n + \gamma(1)A^T A]^{-1}.$$

We use the identity $\|cM\|_2 = |c| \|M\|_2$, where $c$ is any real number, to get

$$\|E_1\|_2 = (1 - \bar{\alpha}_1) \left\| [\mathbf{I}_n + \gamma(1)A^T A]^{-1} \right\|_2.$$

Here, note that $(1 - \bar{\alpha}_1) \geq 0$. Similar to **Lemma C.8**, we can rewrite the spectral norm as

$$\|E_1\|_2 = \frac{1 - \bar{\alpha}_1}{\lambda_{\min}([\mathbf{I}_n + \gamma(1)A^T A])}.$$

Again, using Weyl's inequality and performing similar modifications as in **Lemma C.8**, we obtain the following upper bound for the spectral norm

$$\|E_1\|_2 \leq \frac{1 - \bar{\alpha}_1}{1 + \gamma(1)\lambda_{\min}(A^T A)}.$$

$\square$

**Lemma C.10.** *Suppose **Assumption 4.3** holds. Consider a $T$-step diffusion process with coefficients $\bar{\alpha}_0, \ldots, \bar{\alpha}_T$ such that $\bar{\alpha}_0 = 1$, $\bar{\alpha}_T = 0$, $\bar{\alpha}_t \in [0, 1]$. If $\bar{\alpha}_t < \bar{\alpha}_{t-1} \ \forall \ t \in [0, T]$, $\|F_t\|_2$, with $F_t$ defined as in Equation 17, is less than 1. Additionally, $\|F_1\|_2$ is 0.*

*Proof.* Note that $F_t$ is given by the expression,

$$F_t = \sqrt{1 - \bar{\alpha}_{t-1}}\sqrt{1 - \bar{\alpha}_t}\sqrt{\bar{\alpha}_t}\mathbf{I}_n.$$

First, we use the identity $\|cM\|_2 = |c|\|M\|_2$, where $c$ is any real number. Therefore, we need to show

$$\|F_t\|_2 = \sqrt{1 - \bar{\alpha}_{t-1}}\sqrt{1 - \bar{\alpha}_t}\sqrt{\bar{\alpha}_t} \, \|\mathbf{I}_n\|_2 < 1.$$

For the given conditions on $\bar{\alpha}_0, \ldots, \bar{\alpha}_T$, we observe that at least one of the terms in $\sqrt{1 - \bar{\alpha}_{t-1}}\sqrt{1 - \bar{\alpha}_t}\sqrt{\bar{\alpha}_t}$ is always less than 1. Therefore $\|F_t\|_2 < 1$. And, since $\bar{\alpha}_0 = 1$, for $F_1$, we have $\sqrt{1 - \bar{\alpha}_0} = 0$. Therefore, $F_1$ is a null matrix and $\|F_1\|_2 = 0$. $\square$

# D  Inference Time Results

We evaluated our approach for time series samples up to 576 dimensions (e.g., the Air Quality and the Stocks dataset). We have provided the inference time taken to generate samples with up to 66 and 450 constraints for the Air Quality and the Stocks datasets in Table 9. First, we note that the inference latency for CPS is very similar to PDM [11] and PRODIGY [20], as these approaches involve projection steps after each denoising step. We observe that for univariate datasets, like the Traffic dataset, the inference latency for CPS is less than that of Guided-DiffTime. Note that Guided-DiffTime requires backpropagation through the denoiser network. For multivariate datasets like the Air Quality and the Stocks dataset, the inference time for CPS is roughly $2\times$ more than the inference time for Guided-DiffTime. However, Guided-DiffTime has poor sample quality and very high constraint violation magnitudes. For all the datasets, COP has the least inference time. However, COP also suffers heavily from poor sample quality. Table 9 reports inference latency on a single NVIDIA RTX 6000 GPU.

Additionally, for each real-world dataset, we analyzed the effect of the number of constraint categories. As mentioned in Section 5, our experiments involve imposing multiple constraint types, such as *mean*, *argmin*, *argmax*, and *value at location* constraints. **Interestingly, we observed that the inference latency linearly increases with the number of constraint categories for all the real-world datasets.**

| APPROACH | AIR QUALITY | TRAFFIC | STOCKS |
|---|---|---|---|
| GUIDED-DIFFTIME | 14.76±0.36 s | 11.61±0.39 s | 15.24±0.43 s |
| COP-FT | **8.5±3.72** s | **1.27±0.45** s | **11±4.47** s |
| CPS (OURS) | 31.49±0.64 s | 6.99±0.52 s | 35.22±2.01 s |

Table 9: **The projection step in CPS increases the sampling time.** Here, we present the average inference time taken to generate a single sample for all the real-world datasets used in our experiments. The results are shown in seconds, and the inference time is averaged over 10 runs. Though the inference time for COP-FT is very low, the generated samples have poor sample quality.

Furthermore, we note that there are multiple ways to reduce the inference time for CPS, such as:

- Capping the number of update steps in each projection operation (line 5 of **Algorithm 1**) during the initial denoising steps when the signal-to-noise ratio is very low.

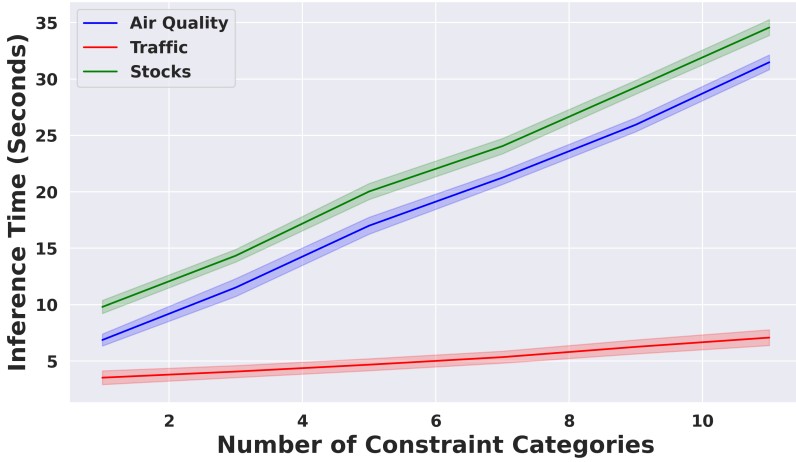

Figure 8: **The inference latency increases linearly with the number of constraint categories.** For all the real-world datasets used in our experiments, we observe that the relationship between the number of constraint categories and the inference time is linear. The experiments are run on a single NVIDIA RTX 6000 GPU. The Traffic dataset has the least inference latency as it has the smallest data dimensionality. The Air Quality and Stocks datasets have the same dimensionality, but the Stocks dataset has an additional OHLC constraint.

- The projection operation (line 5 of **Algorithm 1**) need not be performed for every denoising step. Consequently, we can develop principled methods to identify the denoising steps where projection is required based on constraint violation.

Now, we provide a brief convergence analysis for **Algorithm 1** with regard to **Assumptions 4.1** and **4.3**.

Under **Assumption 4.1**, the objective in Line 5 has $\beta$-Lipschitz continuous gradients, yielding a convergence rate of $\mathcal{O}(1/N_{pr})$, where $N_{pr}$ is the number of projection steps and is in the order of $\mathcal{O}(1/\delta)$ with $\delta$ being the tolerance limit for constraint violation. This repeats for all $T$ diffusion steps.

Under **Assumption 4.3**, line 5 of **Algorithm 1** has a closed-form solution (**Lemma C.2**), and the convergence rate of CPS with respect to $T$ is $\mathcal{O}(1/T)$.

Additionally, from Figure 8, observe that the inference time for the Air Quality and the Stocks datasets is much higher than the inference time for the Traffic dataset. This can be primarily attributed to the dimensionality of the samples in the respective datasets. The sample dimension in the Air Quality and the Stocks datasets is 576 (6 channels with 96 timestamps in each channel), whereas the sample dimension in the Traffic dataset is 96 (1 channel with 96 timestamps).

# E  Additional Qualitative Results

In this section, we provide additional qualitative comparisons between CPS and other baselines.

# F  Metrics

For the FTSD and J-FTSD metrics, we train the time series and condition encoders using the procedure given in [1]. For FTSD, we only train the time series encoder using supervised contrastive loss to maximize the similarity of time series chunks that belong to the same sample. For J-FTSD, we perform contrastive learning training in a CLIP-like manner to maximize the similarity between time series and corresponding paired metadata, as explained in [1]. We use Informer models as the encoders. Additionally, just as in the case of [1, 43], we observe that the approaches corresponding to the lowest values of FD metrics have the lowest TSTR and DTW scores. This further validates the correctness of the FTSD and J-FTSD metrics used for evaluation. To train these models, we used the

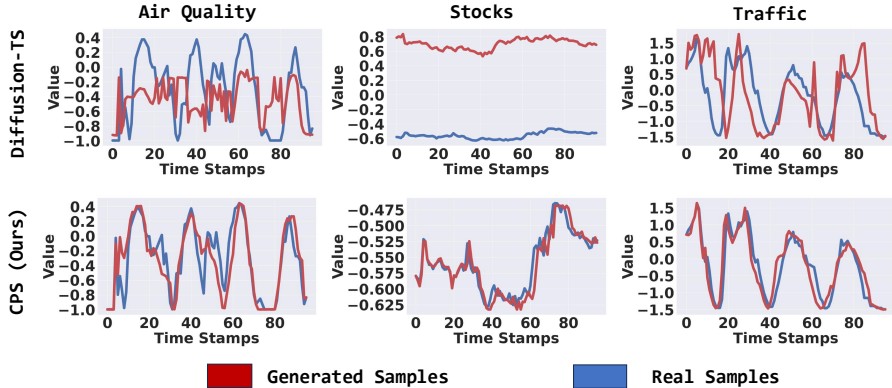

Figure 9: **Qualitative comparison between Diffusion-TS [22] and CPS on the real-world datasets used in our experiments.**

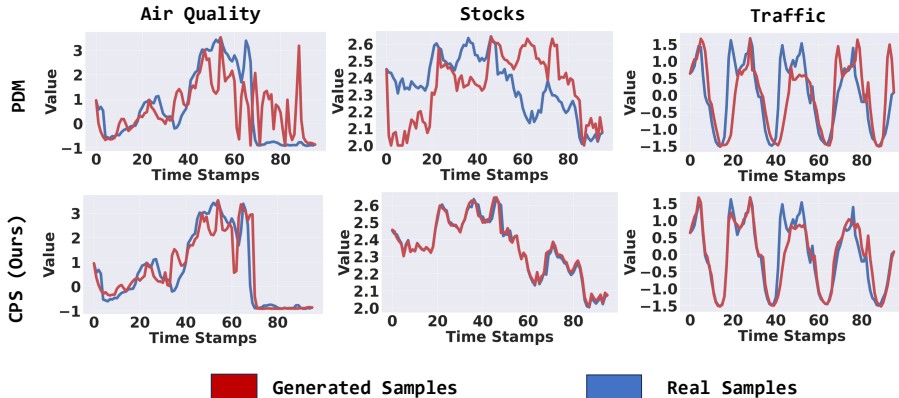

Figure 10: **Qualitative comparison between Projected Diffusion Model (PDM) [11] and CPS on the real-world datasets used in our experiments.**

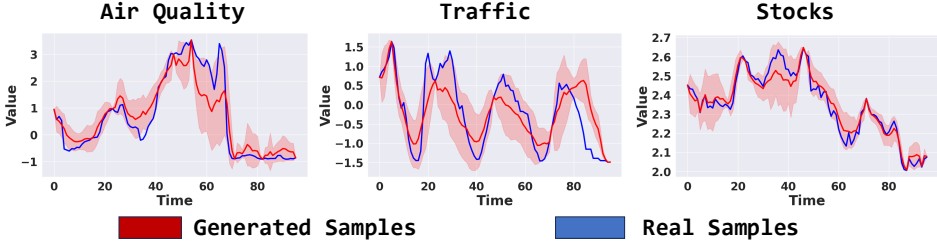

Figure 11: **CPS can generate multiple samples that adhere to the required constraint set.** Here, we showcase CPS's ability to generate multiple samples for the same set of constraints. The real sample from which the constraints are extracted is shown in blue, and we show the mean and $\pm$ standard deviation for the generated samples. Note that the trend of the generated samples matches that of the real sample, and this qualitative result is consistent with other qualitative results in Figures 4, 10, and 9. From the figure, note that the standard deviation at 0, 24, 48, 72, and 96 timestamps goes to zero, as we have imposed the "value at" constraint at these locations.

same set of hyperparameters as mentioned in [1]. For all the real-world datasets, we trained these models up to a maximum of 5000 epochs.

We sourced the implementation for the DTW metric from the public domain. For the constraint violation magnitude, we computed the violation for each constraint, excluding the allowable constraint violation budget. For TSTR, we trained the standard TimesNet [57] model to perform imputation. The mean and standard deviation for the TSTR values are obtained from the results for 3 seeds. We also provide a sample quality comparison based on the Discriminative Score (DS) metric. For this metric, we train a post-hoc time series classification model to distinguish between real and generated

time series samples. We use a simple 2-layer LSTM network for the classification task. DS was introduced in [7] as a sample quality metric. Similar to the TSTR metric, we train the classifier on synthesized and real training data. We then report the classification error on the synthesized and real test data for 5 seeds.

## G   Datasets

We compared CPS against the existing baselines for six settings - Air Quality, Air Quality Conditional, Traffic, Traffic Conditional, Stocks, and Waveforms. The training and testing splits for the Air Quality and Traffic datasets are taken from [1]. We additionally evaluate the constrained generation approaches on the Stocks and the Waveforms datasets. We used the preprocessing scripts provided by [7] for the Stocks dataset. The waveforms dataset was synthetically generated. We generated $64,000$ sinusoidal waveforms of varying amplitudes, phases, and frequencies. The amplitude varies from $0.1$ to $1.0$. The phase varies from $0$ to $2\pi$. The frequency limits were chosen based on the Nyquist criterion. The generators and the GAN models were trained on this dataset. However, for the TSTR metrics, we created a subset of this dataset with $16,000$ samples. All the datasets except the waveforms dataset were standard normalized.

The Air Quality dataset is a multivariate dataset with six channels. The total numbers of train, val, and test samples are 12166, 1537, and 1525, respectively. The Traffic dataset is univariate. The total train, val, and test samples are 1604, 200, and 201, respectively. The Stocks dataset is a multivariate dataset with six channels. The total train, val, and test samples are 2871, 358, and 360, respectively. The truncated form of the waveforms dataset used for evaluation consists of 13320, 1665, and 1665 train, val, and test samples, respectively.

For the conditional variants, we used the same contextual metadata as provided in [1]. For the Air Quality dataset, we used categorical features such as station ID, timestamps, and wind directions, and continuous features such as temperature, pressure, rain levels, and wind speed. For the traffic dataset, we used broad and fine weather descriptions, holidays, and timestamps as categorical features. Similarly, we used the temperature, rain, and snowfall levels, and cloud conditions as continuous features.

## H   Implementation

In this section, we will describe the implementation details for our approach, each baseline, trained models, metrics, etc.

### H.1   Diffusion Model Architecture

We utilize the TIME WEAVER-CSDI denoiser for all the diffusion models used in this work. The training hyperparameters and the model parameters are precisely the same as indicated in [1]. The total number of residual layers is 10 for all the experiments. Further, we used 200 denoising steps with a linear noise schedule for the diffusion process. All the baselines and CPS use the same base diffusion model with the TIME WEAVER-CSDI denoiser backbone.

We use 256 channels in each residual layer, with 16-dimensional vectors representing each channel. The diffusion time step input embedding is a 256-dimensional vector. Further, the metadata encoder has an embedding size of 256 for the conditional case. The metadata encoder has two attention layers with eight attention heads. All our experiments use a learning rate of $10^{-4}$. Our training procedure and the hyperparameters are precisely the same as those in [1]. For each dataset, we trained the diffusion model on a single NVIDIA RTX 6000 GPU. The checkpoints were obtained using the best validation denoising loss, and we trained the TIME WEAVER-CSDI denoiser up to a maximum of 5000 epochs for all datasets.

### H.2   Constrained Posterior Sampling Implementation

For the CPS implementation, we use CVXPY [35]. We first implement the constraint violation function with the violation threshold set to $0.005$ for all the constraints except the bounds, like argmax, argmin, OHLC, and the trend constraint. For example, consider the mean constraint. The constraint

violation function for this constraint is implemented as $\max\left(\left|\frac{1}{L}\left(\sum_{u=1}^{L} c_u\right) - \mu_c\right| - 0.005, 0\right)$, where $L$ is the time series horizon. We do not provide the constraint violation threshold for the bounds. Though the allowable constraint violation threshold is $0.01$, we performed the projection step with a constraint violation threshold of $0.005$ to ensure that the sample strictly lies within the constraint set. We use the same choice of $\gamma(t) \ \forall t \in [1, T]$ as described in Section 3. However, we clip the value of $\gamma(t)$ to $100,000$ after certain denoising steps, as the CVXPY solvers cannot handle extremely high values of $\gamma(t)$. We note that this clipping usually occurs after 150 denoising steps.

To generate samples on a large scale for the training, validation, and test datasets, we performed batched denoising. To parallelize the projection step with CVXPY after the denoising step, we used multiprocessing with 4 processes.

### H.3  Baseline Implementation

This section will explain all the details about the baseline implementations. Specifically, we use two baselines - Constrained Optimization Problem (COP) and Guided DiffTime. We note that both approaches were proposed in [12]. However, the implementation of these approaches is not publicly available. Based on the details provided in [12], we have implemented the baselines for comparison against CPS.

#### H.3.1  Constrained Optimization Problem Implementation

The Constrained Optimization Problem, COP, has two variants. These are referred to as COP and COP-FineTuning, respectively. In COP, we perturb a randomly selected sample from the training and validation datasets. In COP-FineTuning, we perturb the sample generated from the TIME WEAVER-CSDI diffusion model.

Note that [12] suggests extracting statistical features to be imposed as distributional constraints. For example, [12] suggests extracting autocorrelation features for the stocks dataset. However, since it is practically impossible to list all the statistical features for each dataset to obtain the distributional constraints, [12] suggests the use of the critic function from a Wasserstein GAN [21]. The details of the GAN training are summarized below.

COP has two objectives - maximize the $l_2$ distance from a randomly selected sample from the training and maximize the critic value from a Wasserstein GAN.

Similarly, COP FineTuning has two objectives - minimize the $l_2$ distance from a generated sample and maximize the critic value from a Wasserstein GAN.

We optimize for these objectives while ensuring constraint satisfaction.

As suggested in [12], we use the SLSQP solver from SciPy [58]. Unlike [12], which performs piecewise optimization, we note that all the constraints used in our work are global. Therefore, piecewise optimization is very suboptimal. For example, it is suboptimal to break a time series into chunks and perform optimization for each piece when the goal is to generate a sample with a specific mean value. This is also pointed out in [12]. Therefore, we perform COP for the whole time series at once. We consider two budgets - $0.005$ and $0.01$. This is similar to [12]. However, unlike their approach, we stop with $0.01$ as the allowable constraint violation in our case is $0.01$ for all methods.

We used a weight of $0.1$ for the critic's objective. We noticed that for different values $(1.0, 0.1, 0.01)$ of this weight, there was very little change in the DTW metric.

#### H.3.2  Critic Function Implementation

[12] suggests using the critic function in a Wasserstein GAN [21] to enforce realism in the COP approach. Therefore, we used the WaveGAN [8] implementation from [6]. The implementation from [6] has the gradient penalty loss, an improved training procedure to enforce the required Lipschitz continuity for the critic function. Additionally, the WaveGAN training with gradient penalty has been implemented [6] for generating time series samples for the ECG domain. Therefore, we use their implementation to obtain the critic function for the COP baseline. The number of parameters is adjusted such that the diffusion model and the GAN model have a comparable number of parameters.

Similar to the diffusion model, we used the same architecture and training hyperparameters for all the datasets and experimental settings. Specifically, we trained the WaveGAN model with a learning rate of $10^{-4}$ for all the datasets. The input to the generator is a 48-dimensional random vector. Additionally, we ensured that the total number of parameters was equally distributed between the generator and the discriminator to prevent either of the models from overpowering the other. The WaveGAN model was trained for 5000 epochs for all the datasets. For the conditional variants, we restricted the training to 1000 epochs as the training was highly unstable.

### H.3.3 Guided DiffTime Implementation

We use the same TIME WEAVER-CSDI denoiser as in the case of CPS. For the guidance weight, we experimented with the following weights - $(0.00001, 0.0001, 0.001, 0.01, 0.1, 1.0)$. We chose the best guidance weight based on the constraint violation magnitude. Note that we used the same guidance weight for all individual constraints. Using PyTorch, we implemented all the constraints mentioned in Section 5. Additionally, we augmented the Guided DiffTime approach with the DiffTime algorithm for fixed values. In other words, after each step of denoising followed by guidance update, we enforced the fixed value constraints, as specified in [12]. This applies to the values at argmax, argmin, 1, 24, 48, 72, and 96 timestamps.

### H.3.4 Diffusion-TS, PDM and PRODIGY Implementations

For the PRODIGY baseline [20], we used the diffusion step-based coefficient, as the distance-based coefficient does not guarantee constraint satisfaction, even for convex constraints. For the diffusion step-based coefficient, we experimented with $\gamma_0 = 0$ and $p = [0, 1, 5]$. Note that for $p = 0$, we obtain the PDM baseline [11]. For both PRODIGY and PDM baselines, we used the same CVXPY solvers for projection, similar to CPS. For the Diffusion-TS [22], we obtain guidance from the constraint violation. Similar to the Guided-Difftime baseline, we experimented with scaling the guidance by $(0.00001, 0.0001, 0.001, 0.01, 0.1, 1.0)$.

## I Broader Societal Impact

This paper presents Constrained Posterior Sampling (CPS), which is a novel algorithm that focuses on constrained time series sample generation. More broadly, CPS falls under the category of targeted sample generation. CPS has a lot of positive impacts in the time series domain, as it can help in generating targeted samples for stress-testing machine learning systems, as well as replacing private user or enterprise data with more accurate synthetic variants. As such, the societal consequences of our work are limited to those arising from improved quality of generated synthetic data.

