# OpenReview forum: "Constrained Posterior Sampling: Time Series Generation with Hard Constraints"
_NeurIPS.cc/2025/Conference — NeurIPS 2025 poster_

### Official Review · Reviewer_iu3N · 2025-06-24

**Clarity:** 3
**Significance:** 2
**Originality:** 2
**Rating:** 4
**Confidence:** 3

**Summary:**

The paper proposes Constrained Posterior Sampling (CPS) to enable constrained time series generation. The method is based on the DDIM framework but enforces constraints via projections and solves an unconstrained problem in each denoising step. By defining a time-varying penalty approaching infinity, the constraints are enforced for the final steps. CPS does not require specific training and can be applied during inference. Further, unlike guidance-based approaches, it does not require additional hyperparameter tuning but uses off-the-shelf solvers.

**Questions:**

- In L162-166, the paper claims that the algorithm maximizes the likelihood of the generated samples. Can you elaborate on what this means?
- In Figures 4 and 9, the generated samples almost perfectly align with the real samples. Why do multiple samples, i.e., samples with standard deviations, look worse (see Figure 10)? There are multiple timesteps where the real samples are not within the confidence interval.
- Can CPS be extended to non-linear constraints?
- Can you perform imputation using CPS by constraining values to match observations?

I am willing to increase my score if the weaknesses are addressed and if my questions are answered satisfactorily.

**Ethical Concerns:**

["NO or VERY MINOR ethics concerns only"]

**Final Justification:**

The paper proposes an interesting idea and showcases its applicability in various experiments. My concerns have mostly been addressed in the rebuttal. Demonstrating imputation and/or forecasting in small experiments would have further improved the contribution.

**Limitations:**

Yes, the limitations are addressed.

**Quality:**

3

**Strengths And Weaknesses:**

Overall, I like the paper, and it makes valuable contributions. The major weaknesses I see are the simplicity of the method, increased runtime (see Appendix D), and unclear experimental setup. In detail:


### Strengths

- Unlike other approaches, CPS enforces constraints in the data space, i.e., at $t=0$. This seems a more natural approach and allows more flexibility in the latent steps, i.e., $t>0$, and helps to stay in the latent manifolds during generation (L206-222)
- CPS does not require backpropagation through the network. This might be useful in specific settings, i.e., black box settings or non-differentiable constraints.
- The approach is theoretically motivated and analyzed (Sec. 4)
- Experiments with various constraints (e.g., mean, max, min, specific values, ...). Setting specific values is an interesting setup, especially as it should include forecasting and imputation as a special case.
- The background and methodology are mostly clear, and the notation is introduced adequately. In general, the paper is nicely written.

### Weaknesses

- The paper only investigates linear constraints.
- There are some claims lacking empirical evidence. E.g., "projecting the noisy intermediate latents (PDM and PRODIGY) pushes them off the noise manifold" and "This approach provides noisy latents that are closer to the training domain of the denoiser"
- It is unclear which constraints are used in the experiments. L286-287 mentions: "mean, mean consecutive change, argmax, argmin, value at argmax, value at argmin, values at timestamps 1, 24, 48, 72, & 96". Which of these are omitted in Figure 5? Shouldn't the stocks dataset have more constraints? Why is the x-axis labeled "Number of constraints" while Figure 7 in the appendix is labeled "Number of constraint categories"? I assume that the constraints apply to each dimension for multivariate datasets. As this is crucial information to understand the empirical results, this should be clarified.
- "Optimizing the guidance weights seems practically hard for a large number of constraints". Why are the guidance-based approaches not included in Figure 5? It is unclear how they compare for few constraints.

### Minors

- Hyperlinks do not work
- Context-FID has three different names across the paper: Context-FID, (Joint) Frechet Time Series Distance, Frechet Distance.
- The extended related work for time series generation (A.1) should be more thorough.

---

> ### Author Rebuttal · Authors · 2025-07-31
>
> We thank the reviewer for the detailed and thoughtful feedback on our work and for acknowledging the effectiveness of CPS as it does not require any backpropagation through the denoiser.
>
> **On extension to non-linear constraints:**
> We have compared CPS against other baseline methods for a general class of constraints. Specifically, we imposed the **Autocorrelation Function (ACF) value at a particular lag** as an equality constraint for the stocks dataset.
>
> We observe that, even though the projection step does not result in the global optimum, the iterating projection and denoising process pushes the generated sample to the constraint set. Additionally, the sequential denoising process removes the adversarial artifacts caused by the projection step. **This provides very low constraint violation metrics for CPS, when compared to other baselines, while maintaining high sample quality (Appendix B.3, Table 4).** We request the reviewer to refer **On extension to non-linear constraints** section of our response to **Reviewer gJxX** for a detailed explanation.
>
> **On the empirical evidence for the claim made in the paper:**
>
> **Claim** - Projecting the noisy intermediate latent (PDM and PRODIGY) has adversarial effects on sample quality.
>
> **Evidence** - Note that all approaches use the same denoiser. The **only conceptual difference** between these approaches and CPS is that **CPS projects the posterior mean estimate**. Consequently, this results in the noisy latent, $z_{t-1}$, being pushed off the manifold for the diffusion step $t-1$, resulting in imperfect denoising. As a result, imperfect denoising causes sample quality degradation which is empirically shown in Table 2.
>
> Section 3 (lines 229–240) outlines two key reasons for the same.
>
> **On performing imputation with CPS:**
> The reviewer is correct. **CPS can be extended to perform forecasting and imputation** by imposing the **value at** constraint to all the timestamps where the true values are available.
>
> **Clarification regarding the likelihood maximization claim:**
> In lines 162-166, we state that the proposed algorithm satisfies the desired constraints while maximizing the likelihood of the generated samples.
>
> In general, diffusion models are trained to maximize the log-likelihood of observing samples from the dataset, i.e., log p(x). Therefore, diffusion models allow for sampling from p(x).
>
> However, in constrained time series generation, the objective is to sample from p(x) subject to certain constraints.
>
> To sample from p(x), **CPS uses a pre-trained diffusion model that was trained to maximize log-likelihood**. This is what we refer to as maximizing the likelihood of generated samples.
>
> Diffusion models sample from p(x) through sequential denoising. Now, to ensure constraint satisfaction, in CPS, we perturb the denoised sample, after every denoising update, using a projection step towards the constraint set.
>
> This ensures that the generated samples have high likelihood while belonging to the constraint set.
>
> **Performance of Guidance-Based approaches on fewer constraints:**
> In Figure 5, we compare all approaches that can provide satisfaction guarantees for convex constraints. Below, we extend Figure 5 with guidance-based approaches for the Stocks dataset.
>
> | Metric | Approach           | 1 constraint | 3 constraints | 5 constraints | 7 constraints | 9 constraints | 11 constraints |
> |--------|--------------------|--------------|----------------|----------------|----------------|----------------|-----------------|
> | DTW    | Diffusion-TS       | 14.4         | 8.9            | 9.3            | 8.5            | 7.6            | 7.4             |
> |        | Guided Difftime    | 14.8         | 11.7           | 10.1           | 8.3            | 8.1            | 7.9             |
> |        | **CPS**            | **12.5**     | **0.9**        | **0.4**        | **0.3**        | **0.2**        | **0.2**         |
> | FTSD   | Diffusion-TS       | 1.93         | 1.03           | 1.13           | 1.14           | 1.07           | 1.13            |
> |        | Guided Difftime    | 1.56         | 1.59           | 1.39           | 1.25           | 1.12           | 1.24            |
> |        | **CPS**            | **0.94**     | **0.08**       | **0.003**      | **0.002**      | **0.002**      | **0.002**       |
>
> From the table, **CPS comfortably outperforms guidance-based approaches in sample quality**, even for a smaller number of constraints. Additionally, due to the absence of any principled projection step, these approaches do not provide constraint satisfaction even for convex constraints. We observe similar trends for all real-world datasets.
>
> **Clarification regarding the qualitative results in Figures 4, 9, and 10:**
> In Figures 4 and 9, we observe almost perfect overlap for a few datasets, specifically for the conditional variants and stocks. **This is primarily due to conditional inputs and a large number of constraints.**
>
> For the conditional variants (Air Quality Conditional and Traffic Conditional), **the generated samples are already condition or metadata-specific and therefore track the real sample closely.** In addition, when we impose constraints, the generated samples almost perfectly match the real samples. This does not happen in the unconditional cases and is empirically demonstrated through the **differences in the DTW metrics** between the unconditional and conditional cases (2.35 vs 1.83 for Air Quality and 3.41 vs 0.84 for Traffic, lower is better).
>
> For the stocks dataset, the number of constraints is too high (450) that **the feasible set is very small** and therefore the generated samples closely track the real sample from the constraint set.
>
> Consequently, in both these figures, note that the least similarity between the real and generated samples can be observed for the unconditional traffic and air quality datasets which do not have a large number of constraints as the stocks dataset.
>
> The generated samples only roughly follow the trend of the real sample, but do not match as perfectly as compared to the conditional variants.
>
> Therefore, in Figure 10, while depicting the distribution of samples for a given constraint set, we observe higher overall variance for the unconditional traffic and air quality setups when compared to the Stocks dataset which has a larger number of constraints.
>
> Additionally, as we impose the **value at** constraints, note that the standard deviation is 0 at 1, 24, 48, 72, and 96 timestamps.
>
> Finally, the mismatch with the real samples do not translate to poor quality samples. Our objective with the qualitative results is to mainly showcase the matching trends with least distortions due to projections. This is confirmed by the high sample quality metrics for CPS.
>
> **Clarification regarding the constraints used in Figures 5 and 7:**
> The reviewer is correct. The x-axis label for both figures 5 and 7 should be “Number of constraint categories”. We will update figure 5 accordingly. In both figures, we use the following convention:
>
> **1 constraint** -  argmax
>
> **3 constraints** - argmax, value at argmax, value at 96
>
> **5 constraints** - argmin, argmax, value at argmax, value at 72 and 96
>
> **7 constraints** - argmin, argmax, value at argmin, value at argmax, value at 48, 72 and 96
>
> **9 constraints** - argmin, argmax, value at argmin, value at argmax, mean, value at 24, 48, 72 and 96
>
> **11 constraints** - argmin, argmax, value at argmin, value at argmax, mean, mean change, value at 1, 24, 48, 72 and 96
>
> We uniformly distributed the value at constraints over these 6 experimental settings. For the stocks dataset, we always impose the OHLC constraint (lines 291 and 292).
>
> Yes, the constraints are applied to every channel in multivariate datasets like Air Quality and Stocks.
>
> For the **air quality dataset (6 channels)**, there are **66 constraints (6 mean + 6 mean change + 6 argmax + 6 argmin + 6 value at argmax + 6 value at argmin + 5x6 value at constraints)**. The same extends to the conditional variants as well.
>
> For the **traffic dataset (1 channel)**, there are **11 constraints** (similar to the air quality dataset). The same extends to the conditional variants as well.
>
> For the **stocks dataset (6 channels)**, there are **450 constraints (66 (similar to air quality) + 4x96 OHLC constraints)**. Note that the OHLC constraints are on an individual time stamp level.
>
> For the **waveforms dataset (1 channel)**, there are **106 constraints (11 (similar to traffic) + 95 trend constraints between successive time stamps)**.
>
> **Modifications to the manuscript**
>
> 1. We will better describe likelihood maximization in line 162 to 166.
>
> 2. We will include more details regarding non-linear constraints in the experiments section.
>
> 3. We will update the manuscript to use uniform notation for the sample quality metric based on the Frechet Distance.
>
> 4. We will include prior works such as [1], [2], [3], [4], [5], etc., and more.
>
> 5. We will address the issue with the hyperlinks.
>
> 6. We will update the x-axis label for Figure 5.
>
> **References**
>
> [1] The Rise of Diffusion Models in Time-Series Forecasting.
>
> [2] A Versatile Diffusion Model for Audio Synthesis.
>
> [3] Non-autoregressive Conditional Diffusion Models for Time Series Prediction.
>
> [4] Generative Time Series Forecasting with Diffusion, Denoise, and Disentanglement.
>
> [5] Self-Guiding Diffusion Models for Probabilistic Time Series Forecasting.

---

> > ### Author Response · Authors · 2025-08-04
> >
> > Dear Reviewer iu3N,
> >
> > We hope our rebuttal addressed the questions and weaknesses pointed out in your review. As we are nearing the end of the discussion phase, we are committed to addressing further questions or concerns about our contributions during the remaining time.
> >
> > Thank you,
> >
> > Authors.

---

> > ### Comment · Reviewer_iu3N · 2025-08-05
> >
> > Thank you for your response. My concerns have mostly been addressed, and I welcome the mentioned modifications to the manuscript.
> >
> > > CPS can be extended to perform forecasting and imputation by imposing the value at constraint to all the timestamps where the true values are available.
> >
> > While out of scope for the rebuttal, showcasing this in a small experiment and comparing against guided diffusion approaches would greatly benefit the paper.
> >
> > I have increased my score accordingly.

---

> > > ### Author Response · Authors · 2025-08-07
> > >
> > > We are glad that our rebuttal has addressed the reviewer's concerns.
> > >
> > > We sincerely appreciate the thoughtful review, which has improved the quality of our manuscript, and the decision to increase the score.
> > >
> > > As suggested by the reviewer, we will add comparisons against guidance-based approaches for the imputation task in the final version of the manuscript.

---

### Official Review · Reviewer_JbhH · 2025-06-26

**Clarity:** 2
**Significance:** 3
**Originality:** 2
**Rating:** 4
**Confidence:** 3

**Summary:**

This paper tackles generating time series that satisfy hard constraints (for applications like stress-testing and synthetic data generation. The authors propose Constrained Posterior Sampling (CPS), a training-free diffusion-based approach.
The key innovation is projecting the posterior mean estimate onto constraint sets after each denoising step, rather than projecting noisy intermediate latents zₜ as in prior methods (PDM, PRODIGY). The algorithm alternates between standard DDIM denoising and constraint projection using optimization solvers, with stronger constraint enforcement in later steps.
The authors provide theoretical analysis under simplified assumptions (Gaussian data, linear constraints) and evaluate against guidance-based methods, post-processing approaches, and other projection methods across stocks, traffic, and air quality datasets.

**Questions:**

Your theory assumes smooth penalties Π(z) = ||fC(z)||₂² but Algorithm 1 uses non-smooth Π(z) = Σ max(0, fcᵢ(z)). Can you either provide analysis for the non-smooth case or explicitly acknowledge this limitation? Also, why use original noise εθ(zₜ,t) with projected ẑ₀,pr instead of updating for consistency?

You test various constraints but lack systematic analysis of which types work well/poorly. Can you provide a constraint taxonomy with performance guidance and design principles for practitioners?

Your "~100 constraints" claim lacks systematic verification. Can you provide scaling curves (inference time/memory vs constraint count), computational complexity analysis, and practical upper limits?

Guidance methods only tested with 6 hyperparameter values and showed high constraint violations. Did you perform sufficient hyperparameter search to ensure fair comparison?

 Theory derives penalty = 2k(T-t+1)/σₘᵢₙ(AᵀA) but implementation uses e^(1/(1-ε̄ₜ)-1). Can you explain this difference and justify your choice?

**Ethical Concerns:**

["NO or VERY MINOR ethics concerns only"]

**Final Justification:**

Based on the response of authors, I am willing to raise the score to weak accept.

**Limitations:**

No. The authors mention computational overhead and provide some mitigation strategies, but fail to address several key limitations identified in this review:

Some missing Limitations:

- Theory-practice disconnect where theoretical guarantees don't apply to the actual algorithm
- Algorithmic inconsistencies (noise term mismatch) that may affect performance
- Limited understanding of which constraint types are suitable for the approach
- Lack of systematic scalability analysis

**Paper Formatting Concerns:**

No.

**Quality:**

3

**Strengths And Weaknesses:**

Strengths

The core insight to project posterior mean estimates rather than noisy latents is novel and well-motivated. This addresses limitations of existing projection-based methods (PDM, PRODIGY) in a principled way.

The approach addresses an important problem with real applications. The training-free nature and elimination of hyperparameter tuning represent practical advantages over existing methods.

The experimental evaluation is sound with appropriate baselines, multiple evaluation metrics, and fair comparisons using the same diffusion backbone. Results show substantial improvements across diverse datasets.

The intuition that projecting clean estimates preserves the noise manifold structure while enforcing constraints is technically strong.

Weaknesses

The theoretical analysis assumes smooth penalty functions (Π(z) = ||fC(z)||₂²) but the algorithm uses non-smooth penalties (Π(z) = Σ max(0, fcᵢ(z))). This fundamental mismatch raises questions about value of the main theoretical contributions. The theory analyzes equality constraints while practice focuses on inequality constraints.

Algorithm 1 uses original noise predictions εθ(zₜ,t) with projected clean estimates ẑ₀,pr, violating the diffusion model relationship zₜ = √ε̄ₜ·z₀ + √(1-ε̄ₜ)·ε. This inconsistency is neither acknowledged nor justified.

The relationship between theoretical guarantees and practical performance is unclear. Theorem 4.2 is essentially tautological, and the restrictive assumptions (Gaussian data, optimal denoiser) limit practical relevance of the convergence analysis.

No systematic evaluation of constraint types, computational scalability beyond limited evidence, or design guidance for practitioners. The claim of handling "~100 constraints" lacks systematic verification.

The penalty schedule derived in theory differs from implementation without explanation.

While the practical contribution is valuable, the theoretical analysis provides little insight due to assumption violations and oversimplified settings that don't apply to the actual algorithm.

Limited hyperparameter search for baseline methods (only 6 guidance weight values) potentially understates competitor performance. Insufficient analysis of constraint diversity and scaling behavior.

The paper presents a valuable practical contribution undermined by theoretical claims that don't apply to the actual method and missing systematic analysis needed for this type of venue.

---

> ### Author Rebuttal · Authors · 2025-07-31
>
> We thank the reviewer for the detailed feedback on our work and for acknowledging the effectiveness of projecting the posterior mean estimate over the existing approaches like PDM and PRODIGY.
>
> **Regarding the update step in Algorithm 1:**
> Combining lines 7-9 in Algorithm 1, our update step is exactly the same as the DDIM update step with $\hat{z_0}$ being replaced by $\hat{z_{0, pr}}$.
>
> The original DDIM update step results in sampling from an unconditional distribution, whereas, updating $\hat{z_0}$ with $\hat{z_{0, pr}}$ can be viewed as sampling conditioned on the specified constraints. [1] has employed similar algorithms for image personalization.
>
> Further, we observed that using $\hat{\epsilon}$ is better for sample quality metrics. Below, we list the FD and the DTW metrics for CPS and CPS with the noise term updated for consistency.
>
>
> | Metrics | Approach                 | Air Quality      | Traffic           | Stocks           | Waveforms        |
> |---------|--------------------------|------------------|-------------------|------------------|------------------|
> | FTSD    | CPS                      | **0.0234**       | **0.2077**        | **0.0023**       | **0.0029**       |
> |         | CPS with noise correction| 0.2085           | 0.5147            | 0.0426           | 0.0694           |
> | DTW     | CPS                      | **2.35 ± 1.48**  | **1.83 ± 1.16**   | **0.20 ± 0.71**  | **0.23 ± 0.17**  |
> |         | CPS with noise correction| 3.07 ± 1.86      | 4.05 ± 1.26       | 0.54 ± 1.10      | 0.46 ± 0.33      |
>
> Updating both posterior mean and noise results in overdependency on the projection step and pushes $z_{t-1}$ off the noise manifold for $t-1$. This takes $z_{t-1}$ far away from the training domain of the denoiser, resulting in poor sample quality as shown by the numbers above.
>
> **On the mismatch in the penalty schedule/coefficients between theory and experiments:**
> Firstly, the exact penalty coefficients as shown in the theorem cannot be computed for real-world applications as constraints need not satisfy Assumption 4.3.
>
> Therefore, our choice of penalty coefficients in Algorithm 1 is similar in essence to the penalty coefficients used in Theorem 4.4. The key takeaway from Theorem 4.4 is that, for convergence, the penalty coefficients should be **decreasing with respect to $t$** (diffusion step) and can be chosen to increase from a small value during the initial denoising steps (t=T) to a very large value ($\gamma(1) \rightarrow \infty$) during the final denoising steps.
>
> We ablated with **linearly, quadratically, and exponentially (lines 183 -187) decreasing** function choices (Table 5 of Appendix B.3 (page 21)). Note that the **linear functional form**, where the penalty coefficient linearly decreases from 10000 to 0 (increases during denoising), **is proportional to the penalty coefficients obtained from Theorem 4.4 at all diffusion steps**.
>
> Interestingly, on all the real-world datasets, our key observation is that **the function choice does not matter**, and we obtain very similar sample quality metrics with linear, quadratic, and exponential choices (difference in third decimal place, check Table 5 of Appendix B.3 (page 21)). The results in the main table 1 are obtained using the exponential choice described in Algorithm 1.
>
> **On the mismatch between penalty functions in Algorithm 1 and in Theorem 4.2:**
> The smoothness assumption on the penalty function for Theorem 4.2 followed by non-smooth penalty functions in the experimental setup and Algorithm 1 follow prior works from posterior sampling for inverse problems [2,3]. Prior works draw theoretical insights from linear and noise-free observation settings and extend them to non-linear and even non-differentiable cases.
>
> Theorem 4.2's main focus is expressing the projection step as a series of gradient updates with fixed step sizes to convert $\hat{z_0}$ into $\hat{z_{0, pr}}$. Consequently, the **projection step can be viewed as sampling from a Dirac delta distribution centered at $\hat{z_{0, pr}}$.**
>
> Here, the smoothness assumption only aids in ease of choosing the step size and the number of gradient updates, as described in Eq. 7 (page 21, Appendix C.1). This reasoning can also be extended to optimizing non-smooth penalty/objective functions in line 5 of Algorithm 1 as $\hat{z_{0, pr}}$ can be expressed as a sequence of updates from $\hat{z_0}$.
>
> We also note that our experiments include both equality (mean, value at, etc.,) and inequality constraints (OHLC).
>
> **On the usefulness of theoretical analysis:**
>
> **Theorem 4.2:** The theorem draws equivalence between the projection step and sampling from a Dirac delta distribution centered at $\hat{z_{0, pr}}$. Essentially, the theorem provides the modified $p(z_{t-1} | z_t)$ under smoothness assumptions on the penalty function. As described earlier, these insights can be extended to non-smooth cases.
>
> **Theorem 4.4:** Theorem 4.4 investigates the efficacy of CPS on the well-studied linear problem with unique solution and shows that we **can do away with costly constrained optimization** after each denoising step (baseline approaches - PDM and PRODIGY) and perform cheaper unconstrained optimization with steadily increasing penalty for constraint violation as we denoise.
>
> Secondly, the theorem **provides a framework for the choice of penalty coefficients** (small during initial denoising steps and very large during the final denoising steps). **This works well with our intuition that constraint satisfaction need not be heavily prioritized during the initial steps when the signal-to-noise ratio in the denoised sample is very low.**
>
> We also note that CPS, designed based this intuition, outperforms state-of-the-art approaches on sample quality.
>
> **Regarding the hyperparameter search for guidance-based methods:**
>
> Below, we provide a detailed analysis of how the constraint violation magnitude varies with the guidance weights.
> | Approach        | 0.00001 | 0.00005 | 0.0001 | 0.0005 | 0.001  | 0.005  | 0.01  | 0.05    |
> |----------------|---------|---------|--------|--------|--------|--------|-------|---------|
> | Guided Difftime| 23.21   | 23.94   | 25.37  | 51.27  | 113.31 | 951.62 | 2929  | 116084  |
> | DiffTS         | 5.61    | 5.67    | 5.68   | 5.68   | 5.7    | 5.67   | 5.64  | 5.67    |
>
> Note that, for Guided DiffTime, as the guidance weight increases, the violation magnitude also increases drastically.
>
> Whereas, for DiffTS, the violation magnitude largely remains the same with minor fluctuations.
>
> For fair comparison, the guidance weight corresponding to the smallest violation magnitude from this table is used in our experiments.
>
> **On the inference latency for large number of constraints (~100):**
> We refer the reviewer to Appendix D (pages 34 and 35) for a detailed analysis on inference latency. Our key observation is that the **inference latency linearly increases with the number of constraint categories** for all the real-world datasets.
>
> In Figures 5 (page 9) and 7 (page 34), and Table 6 (page 34), we increase the number of constraint categories and observe the sample quality and inference latency.
>
> We request the reviewer to check the **Clarification regarding the constraints used in Figures 5 and 7** in our response to **Reviewer iu3N** for the list of constraints used in each experimental setup. In the caption for Figure 7, we also provide details regarding the hardware setup used for our experiments.
>
> From Table 6, note that the average latency for the univariate traffic dataset is small, when compared to the other multivariate  datasets (Air Quality and Stocks), for any number of constraint categories.
>
> Additionally, note that the Stocks dataset has higher latency for any number of constraint categories because of the large number of OHLC constraints being imposed.
>
> We also provide convergence analysis for Algorithm 1 under both Assumptions 4.1 and 4.3 in lines 1197 to 1201.
>
> **On the systematic evaluation of individual constraint types:**
>
> Below, we provide the FTSD (sample quality) comparison for individual constraint types evaluated on the traffic dataset.
>
> | Approach         | argmax | argmin | Value at argmax  | Value at argmin  | mean | Mean change | Value at (1,24,48,72,96) |
> |------------------|--------|--------|------|------|------|--------------|-----------|
> | Guided Difftime  | 0.21   | 0.22   | 0.21 | 0.3  | 0.22 | 0.22         | 0.24      |
> | COP-FT           | 0.27   | 0.25   | 0.2  | 0.26 | 0.28 | **0.19**     | 0.23      |
> | PDM              | 0.21   | **0.20** | **0.19** | 0.23 | 0.19 | 0.20         | 0.20      |
> | **CPS**          | **0.20** | 0.21   | **0.19** | **0.21** | **0.18** | 0.20         | **0.19**   |
>
> From the table, we note that **even for individual constraint types, CPS outperforms the baselines on sample quality for a large number of constraint types**. Further, while combining multiple constraints together, from Figure 5 (page 9), we clearly observe that CPS provides the best sample quality for any number of constraints.
>
> **Modification to the manuscript**
>
> 1. We will add the tables for systematic verification of constraints and hyper parameter search to the paper.
>
> 2. In the experiments and the limitations sections, we will address the differences in the penalty functions and coefficients between the theoretical justification section and Algorithm 1.
>
> 3. We will include a discussion and the table regarding consistency in the update step.
>
> 4. We will include a paragraph at the end of the theoretical justification  section to explain the usefulness of the theorems.
>
> **References**
>
> [1] RB-Modulation: Training-Free Personalization of Diffusion Models using Stochastic Optimal Control
>
> [2] Pseudoinverse-Guided Diffusion Models for Inverse Problems
>
> [3] Beyond First-Order Tweedie: Solving Inverse Problems using Latent Diffusion

---

> > ### Author Response · Authors · 2025-08-04
> >
> > Dear Reviewer JbhH,
> >
> > We hope our rebuttal addressed the questions and weaknesses pointed out in your review. As we are nearing the end of the discussion phase, we are committed to addressing further questions or concerns about our contributions during the remaining time.
> >
> > Thank you,
> >
> > Authors.

---

> ### Comment · Reviewer_JbhH · 2025-08-05
>
> I appreciate the authors' detailed response and the additional empirical evidence provided. However, after careful consideration, I am maintaining my original rating. While the authors have addressed several technical questions and provided valuable empirical analysis, the core methodological concerns that motivated my original assessment remain unresolved. My main issue is not the absence of perfect theory, but rather the presentation of theoretical contributions that don't apply to the actual method.

---

> > ### Author Response · Authors · 2025-08-07
> >
> > We are glad that our rebuttal answered the reviewer’s queries regarding systematic analysis of individual constraints, proper evaluation of guidance-based methods, and updating the noise estimate for consistency.
> >
> > Below, we provide the reasons why the theoretical justifications aid and improve the understanding of our proposed algorithm, CPS.
> >
> > **Assumption 4.1 (Smoothness assumption on the penalty function) and Theorem 4.2**
> >
> > **Why do we use this assumption?**
> > The smoothness assumption allows for using well-established literature to determine the number of steps and the required step size for the unconstrained optimization step (line 5) in Algorithm 1.
> >
> > **Prior works with similar assumptions for theory, including extension to practical applications:**
> > Our assumption follows prior works on posterior sampling for inverse problems in the image domain [1,2].
> >
> > [1] uses linear inverse problems without noise to obtain the reverse sampling step and replaces the pseudoinverse of the measurement operator with non-linear and non-differentiable operations for reconstructing images compressed using JPEG encoding.
> >
> > [2] provides theoretical guarantees for the proposed sampling algorithms using a linear setting and extends to complex image editing tasks.
> >
> > **Implications of the theorem:**
> > The theorem draws equivalence between the projection step and sampling from a Dirac delta distribution centered at $\hat{z_{0,pr}}$. Prior constrained generation works with diffusion models (PDM and PRODIGY) do not provide this equivalence, which helps in obtaining the modified $p(z_{t-1}|z_t)$ due to the projection step.
> >
> > **How does the proposed theorem apply to CPS and our experimental setup?**
> > Theorem 4.2 can be extended to non-smooth penalty functions as in Algorithm 1 because the projection operation (line 5) can be written as a series of updates from $\hat{z_0}$ to $\hat{z_{0,pr}}$. Therefore, Theorem 4.2 still provides the updated $p(z_{t-1}|z_t)$ for Algorithm 1.
> >
> > **Assumption 4.3 and Theorem 4.4**
> >
> > **Why is this assumption required?**
> > The rank assumption provides a unique sample that needs to be generated. This helps with the convergence analysis. The Gaussian data distribution assumption allows for the usage of an optimal denoiser. This helps in eradicating convoluting factors caused by improper denoising and allows us to study the biases introduced by Algorithm 1.
> >
> > **Prior works with similar assumptions for theory:**
> > We note that similar assumptions (unique solution) are made in the theoretical analysis of algorithms designed for sample recovery and image inpainting [3,4] using diffusion models.
> >
> > **Implications of the theorem:**
> > Theorem 4.4 provides a penalty schedule that ensures convergence within the tolerance limit. Overall, the penalty coefficient linearly increases from 0 to a very large value as we denoise.
> >
> > **How does the proposed theorem apply to CPS?**
> >
> > Not all real-world constraints satisfy Assumption 4.3.
> >
> > Therefore, the penalty coefficients in Algorithm 1 (exponentially increases from 0 to a very large value as we denoise) are designed similar in essence to the penalty coefficients used in Theorem 4.4.
> >
> > Moreover, we ablated with linearly, quadratically, and exponentially (lines 183 -187) increasing penalties (Table 5 of Appendix B.3 (page 21)) and find that empirically the sample quality is unaffected by the functional form of the penalty coefficients.
> >
> > The linearly increasing penalty coefficients are proportional to the penalty coefficients obtained from Theorem 4.4 at all diffusion steps.
> >
> > We hope our response provides the necessary connection between the theorems and our proposed algorithm along with its practical applications.
> >
> > We will include a section following the theoretical justification section to explain how the theorems contribute to a better understanding of CPS and how they can be extended to practical real-world constraints.
> >
> > **References**
> >
> > [1] Pseudoinverse-Guided Diffusion Models for Inverse Problems
> >
> > [2] Beyond First-Order Tweedie: Solving Inverse Problems using Latent Diffusion
> >
> > [3] Score approximation, estimation and distribution recovery of diffusion models on low-dimensional data
> >
> > [4] A Theoretical Justification for Image Inpainting using Denoising Diﬀusion Probabilistic Models

---

### Official Review · Reviewer_gJxX · 2025-06-30

**Clarity:** 3
**Significance:** 2
**Originality:** 2
**Rating:** 4
**Confidence:** 3

**Summary:**

The paper presents a new diffusion-based algorithm for generating time series satisfying linear constraints. Without retraining the denoiser, it only alters the reverse sampler by enforcing all constraints on the posterior mean at every denoising step. Compared to previous methods, it better preserves sample quality while satisfying the constraints.

**Questions:**

Why are the generated series matching a real time series pointwise almost exactly?

**Ethical Concerns:**

["NO or VERY MINOR ethics concerns only"]

**Limitations:**

yes

**Paper Formatting Concerns:**

None.

**Quality:**

3

**Strengths And Weaknesses:**

**Strengths**

- The paper is clear and well-written.
- The method’s rationale is well explained and supported by empirical results.
- The empirical evaluation is sound and extensive: compares to representatives of different paradigms (guidance, post-processing, and Projected Diffusion Models (PDM)); uses multiple metrics, visualises samples, and measures cost and performance versus the number of constraints.
- The proposed method is relatively easy to implement. It lets practitioners impose multiple convex constraints with solid theoretical guarantees in the linear case.

**Weaknesses**

- Limited novelty: the main difference from PDM is projecting the posterior mean instead of the latent state.
- Narrow scope: focuses on convex constraints. The appendix’s non-convex autocorrelation example (fixed lag $k$) is interesting but its discussion (lines 852–853) is too vague.

---

> ### Author Rebuttal · Authors · 2025-07-31
>
> We thank the reviewer for the feedback on our work. The reviewer acknowledges the depth of the provided empirical evaluations along with the simplicity and the theoretical guarantees for CPS.
>
> **On the range of constraints used in experiments:** We note that the prior work [1] in constrained time series generation primarily focused only on linear or convex constraints individually (argmin, fixed value, and bounds). The approaches in [1] have not been tested for various combinations of constraints on multiple real-world datasets.
>
> However, our experiments involve detailed evaluation on multiple combinations of constraints using a range of metrics that capture the sample quality and diversity.
>
> Additionally, the number of constraints imposed on a generated sample is much higher (~100 and upto 450 for the stocks dataset) in our work than in [1]. We request the reviewer to check the **Clarification regarding the constraints used in Figures 5 and 7** in our response to **Reviewer iu3N.**
>
> **On extension to non-linear constraints:**
> We have compared CPS against other baseline methods for a general class of constraints. Specifically, we imposed the **Autocorrelation Function (ACF) value at a particular lag as an equality constraint** for the stocks dataset.
>
> Firstly note that even CPS has constraint violation in this case as the constraint is non-convex.
> Secondly, the projection step, after every denoising step, does not provide the global minimum for $\hat{z}_{0,\mathrm{pr}}$, which was the case for linear and convex constraints used in the main table 1.
>
> Even in such cases where the projection operation does not yield a $\hat{z}_{0,\mathrm{pr}}$ which is the global minimum, **the iterative projection and denoising steps in CPS ensure that the adverse effects of projection are minimized and provide high sample quality.**
>
> We observe that CPS provides the best sample quality metrics while maintaining the least constraint violation magnitude (Appendix B.3, Table 4).
>
> **On limited novelty:** Though the key difference between CPS and other approaches such as PDM and PRODIGY is the projection of the posterior mean estimate, we note that **CPS applies the constraints on the estimate of a clean sample and not on the noisy latents**, in accordance with the fact that the constraints are defined for clean samples. This improves the sample diversity (lines 215-218).
>
> Additionally, this key difference ensures the modified latents ($z_{t-1}$) lie on the manifold corresponding to the noise level $t-1$, thereby ensuring accurate denoising with respect to PDM and PRODIGY. This results in improved sample quality (lines 219-222).
>
> **Regarding the qualitative results:**
> In Figures 4 and 9, we observe almost perfect overlap between the real and the generated time series samples for a few datasets. Specifically, this can be observed for the conditional variants of the Air Quality and Traffic datasets along with the Stocks dataset.
>
> **This is primarily due to conditional inputs and a large number of constraints.**
>
> For the conditional variants (Air Quality Conditional and Traffic Conditional), the generated samples are already condition or metadata-specific and therefore track the real sample closely. In addition, when we impose constraints, the generated samples almost perfectly match the real samples. This does not happen in the unconditional cases and is empirically demonstrated through the **differences in the DTW metrics between the unconditional and conditional cases** (2.35 vs 1.83 for Air Quality and 3.41 vs 0.84 for Traffic, lower is better).
>
> For the stocks dataset, the number of constraints is too high (450) that the **feasible set is very small** and therefore the generated samples closely track the real sample from the constraint set.
>
> Consequently, in both these figures, note that the least similarity between the real and generated samples can be observed for the unconditional Traffic and Air Quality datasets which do not have a large number of constraints as the Stocks dataset. The generated samples only roughly follow the trend of the real sample, but do not match as perfectly as compared to the conditional variants.
>
> Accordingly, as diffusion models generate stochastic outputs, Figure 10 depicts the distribution of samples generated for the same constraint set. We observe higher overall variance for unconditional Traffic and Air Quality datasets when compared to the Stocks dataset which has a larger number of constraints.
>
> Additionally, as we impose the **value at** constraints, note that the standard deviation is 0 at 1, 24, 48, 72, and 96 timestamps.
>
> Our objective with the qualitative results is to mainly showcase the matching trends with least distortions due to projections. This is also confirmed by the high sample quality metrics for CPS.
>
> **Modifications to the manuscript**
>
> 1. We will include explanation regarding non-linear constraints in the experiments section.
>
> 2. We will add a paragraph explaining the qualitative results and the perfect overlap on selected datasets.
>
> **References**
>
> [1] On the Constrained Time-Series Generation Problem

---

> > ### Author Response · Authors · 2025-08-04
> >
> > Dear Reviewer gJxX,
> >
> > We hope our rebuttal addressed the questions and weaknesses pointed out in your review. As we are nearing the end of the discussion phase, we are committed to addressing further questions or concerns about our contributions during the remaining time.
> >
> > Thank you,
> >
> > Authors.

---

### Note · Authors · 2025-08-15

We thank the reviewers for their thoughtful and thorough feedback on our manuscript. Through our rebuttal and further discussions, we hope we addressed the queries and doubts raised by the reviewers. To this end, we will make the following key changes to our manuscript:

1. We will add the tables for systematic verification of constraints, hyperparameter search, and provide more information on non-linear constraints in the experiments section.

2. We will address the differences in the penalty functions and coefficients between the theoretical justification section and Algorithm 1. Further, we will include an explanation regarding how the theorems aid in understanding CPS better.

3. We will include a discussion and the table regarding consistency in the update step.

4. We will update the manuscript to use uniform notation for the sample quality metric based on the Frechet Distance.

5. We will extend the additional literature review in Appendix A and include imputation experiments.

Thank you,

Authors.

---

### Decision · Program_Chairs · 2025-09-17

**Decision:**

Accept (poster)

**Comment:**

This paper proposes Constrained Posterior Sampling, a diffusion-based method for constrained time series generation.
The reviewers all found the approach novel and well-motivated, with good empirical results and clear writing. Some weakness was also noted, including the theoretical analysis that relies on assumptions that do not fully match the implemented algorithm. Other concerns included clarity on constraint types, scalability analysis, and handling of non-linear constraints. The authors rebuttal helped and the committed to improve the paper in the camera-ready version, was appreciated.

Overall, despite some theoretical weaknesses, the method represents a meaningful and useful contribution.